# Constrained Sampling with Primal-Dual Langevin Monte Carlo

**Luiz F. O. Chamon**
University of Stuttgart
`luiz.chamon@simtech.uni-stuttgart.de`

**Mohammad Reza Karimi**
ETH Zürich
`mkarimi@inf.ethz.ch`

**Anna Korba**
CREST, ENSAE, IP Paris
`anna.korba@ensae.fr`

## Abstract

This work considers the problem of sampling from a probability distribution known up to a normalization constant while satisfying a set of statistical constraints specified by the expected values of general nonlinear functions. This problem finds applications in, e.g., Bayesian inference, where it can constrain moments to evaluate counterfactual scenarios or enforce desiderata such as prediction fairness. Methods developed to handle support constraints, such as those based on mirror maps, barriers, and penalties, are not suited for this task. This work therefore relies on gradient descent-ascent dynamics in Wasserstein space to put forward a discrete-time primal-dual Langevin Monte Carlo algorithm (PD-LMC) that simultaneously constrains the target distribution and samples from it. We analyze the convergence of PD-LMC under standard assumptions on the target distribution and constraints, namely (strong) convexity and log-Sobolev inequalities. To do so, we bring classical optimization arguments for saddle-point algorithms to the geometry of Wasserstein space. We illustrate the relevance and effectiveness of PD-LMC in several applications.

## 1 Introduction

Sampling is a fundamental task in statistics, with applications to estimation and decision making, and of growing interest in machine learning (ML), motivated by the need for uncertainty quantification and its success in generative tasks [1–3]. In these settings, the distribution we wish to sample from (*target distribution*) is often known only up to its normalization constant. This is the case, for instance, of score functions learned from data or posterior distributions of complex Bayesian models. Markov Chain Monte Carlo (MCMC) algorithms can be used to tackle this problem [4, 5] and Langevin Monte Carlo (LMC), in particular, has attracted considerable attention due to its simplicity, theoretical grounding, and effectiveness in practice [3, 6–9]. These sampling algorithms, however, do not naturally incorporate requirements on the samples they generate. Specifically, standard MCMC methods do not enforce restrictions on the target distributions, such as support (e.g., truncated Gaussian), conditional probabilities (e.g., fairness), or moments (e.g., portfolio return) constraints. This limitation is often addressed by post-processing, transforming variables, or by introducing penalties in the target distribution. Though successful in specific settings, these approaches have considerable downsides. Post-processing techniques such as *rejection sampling* (see, e.g., [10, 11]) may substantially reduce the effective number of samples (number of samples generated per iteration of the algorithm). Variable transformations based on link functions, projections, or mirror/proximal maps (see, e.g., [12–18]) only accommodate (deterministic) support constraints and are not suited

Table 1: Type and target of sampling constraints.

| | Soft constraint | Hard constraint |
|---|---|---|
| Sample ($x \sim \mu$) | — | Mirror/proximal LMC [12, 14, 15, 17], projected LMC [13], barriers [16, 18] |
| Distribution ($\mu$) | Penalized LMC [23] | **PD-LMC** |

for statistical requirements such as robustness or fairness [19–22]. Though modifying the target distribution directly offers more flexibility (see, e.g., [23]), it does not *guarantee* that constraints are satisfied (Table 1). We refer the reader to Appendix A for a more detailed literature review.

This paper overcomes these issues by directly tackling the *constrained sampling* problem. Explicitly, it seeks to sample not from a target distribution $\pi$ on $\mathbb{R}^d$, but from the distribution $\mu^\star$ that solves

$$
\begin{aligned}
P^\star \triangleq \min_{\mu \in \mathcal{P}_2(\mathbb{R}^d)} \quad & \mathrm{KL}(\mu \| \pi) \\
\text{subject to} \quad & \mathbb{E}_{x \sim \mu}\big[g_i(x)\big] \leq 0, \ \ i = 1, \ldots, I, \\
& \mathbb{E}_{x \sim \mu}\big[h_j(x)\big] = 0, \ \ j = 1, \ldots, J,
\end{aligned}
\tag{PI}
$$

where $\mathcal{P}_2(\mathbb{R}^d)$ denotes the set of probability measures on $\mathbb{R}^d$ with bounded second moments and the functions $g_i, h_j$ represent the requirements. Note that (PI) only considers measures $\mu$ against which $g_i, h_j$ are integrable. Otherwise, the expectations are taken to be $+\infty$, making the corresponding measure infeasible. Observe that (PI) is more general than the support-constrained sampling problem considered in, e.g., [12–18]. Indeed, it constrains the distribution $\mu$ rather than its samples $x$ (Table 1). Algorithms based on projections, barrier, or mirror maps are not suited for this type of constraints (see Section 2.2 for more details). To tackle (PI), this paper instead derives and analyzes a primal-dual LMC algorithm (PD-LMC) that is the sampling counterpart of gradient descent-ascent (GDA) methods from (Euclidean) optimization. A dual ascent algorithm was previously proposed to tackle (PI), but it requires the exact computation of expectations with respect to intractable distributions [24]. This paper not only overcomes this limitation, but also provides convergence guarantees for a broader class of constraint functions.

The main contributions of this work include:

- a discrete-time constrained sampling algorithm (PD-LMC, Algorithm 1) for solving (PI) that precludes any explicit integration (Section 3);

- an analysis of PD-LMC proving that it converges sublinearly (in expectation) with respect to the Kullback-Leibler divergence (convex case) or Wasserstein distance (strongly convex case). The analysis is performed directly on the discrete-time iterations and requires only local Lipschitz continuity and bounded variance assumptions (Section 3.1);

- an extension of these result for target distributions satisfying a log-Sobolev inequality (LSI) for a variant of PD-LMC (Algorithm 2, Section 3.2);

- numerical examples illustrating applications of (PI) and the effectiveness of PD-LMC (Section 4).

## 2 Problem formulation

### 2.1 Background on Langevin Monte Carlo

Consider a target distribution $\pi \in \mathcal{P}_2(\mathbb{R}^d)$ absolutely continuous with respect to Lebesgue measure whose density (also denoted $\pi$) can be expressed as $\pi(x) = e^{-f(x)}/Z$ for some normalization constant $Z$. Define the Kullback-Leibler (KL) divergence of $\mu$ with respect to $\pi$ as

$$
\mathrm{KL}(\mu \| \pi) = \int \log\left(\frac{d\mu}{d\pi}\right) d\mu = \int f d\mu + \int \log(\mu) d\mu - \log(Z) \triangleq \mathcal{V}(\mu) + \mathcal{H}(\mu) - \log(Z), \tag{1}
$$

where $\frac{d\mu}{d\pi}$ is the Radon–Nikodym derivative, $\mathcal{V}$ is the *potential energy*, and $\mathcal{H}$ is the *negative entropy* if $\mu$ is absolutely continuous with respect to $\pi$; and $+\infty$ otherwise. For a wide class of functions $f$ (e.g.,

smooth and strongly convex), samples from $\pi$ can be obtained from the path of the *Langevin diffusion* process, whose instantaneous values $x(t)$ have distributions $\mu(t)$ evolving according to the *Fokker–Planck equation* [25]. Explicitly,

$$dx(t) = -\nabla f(x(t))dt + \sqrt{2}dW(t) \quad \text{and} \quad \frac{\partial \mu(t)}{\partial t} = \nabla \cdot \left[ \mu(t)\nabla_{W_2}\mathrm{KL}(\mu(t)\|\pi) \right], \tag{2}$$

for a $d$-dimensional Brownian motion $W(t)$, where $\nabla \cdot q$ denotes the divergence of $q$ and $\nabla_{W_2}\mathrm{KL}(\mu\|\pi)$ denotes the Wasserstein-2 gradient of $\mathrm{KL}(\cdot|\pi)$ at $\mu$ [26, Theorem 10.4.17] (see Appendix B for more details). Indeed, the Langevin diffusion (2) brings the distribution $\mu(t)$ of $x(t)$ progressively closer to the target $\pi$. In fact, the Fokker-Planck equation can be interpreted as a gradient flow of the KL divergence with respect to the Wasserstein-2 distance [7, 25].

However, computing the path of the stochastic differential equation in (2) is not practical and discretizations are used instead. Chief among them is the (forward) Euler–Maruyama scheme, which leads to the celebrated Langevin Monte Carlo (LMC) algorithm [6]

$$x_{k+1} = x_k - \gamma_k \nabla f(x_k) + \sqrt{2\gamma_k}\beta_k, \quad \beta_k \overset{\text{iid}}{\sim} \mathcal{N}(0, \mathrm{I}_d), \tag{3}$$

for a step size $\gamma_k > 0$, where $\mathrm{I}_d$ denotes the $d$-dimensional identity matrix. Notice that it is not necessary to know $Z$ in order to evaluate (3). This has made LMC and its variants widely popular in practice and the subject of extensive research. Despite (3) being a biased time-discretizations of the Langevin diffusion in (2) [7], rates of convergence of LMC have been obtained for smooth and strongly convex [8, 27] or convex [28] potentials or when the target distribution $\pi$ verifies an LSI [29].

## 2.2 Constrained sampling

Our goal, however, is not to sample from $\pi$ itself, but from a distribution close to $\pi$ that satisfies a set of statistical requirements. Explicitly, we wish to sample from a distribution $\mu^\star$ that solves (PI). Since (PI) constrains the distribution $\mu$ rather than its samples $x$, it can accommodate more general requirements than the support constraints typically considered in constrained sampling (e.g., [12–18]). Next, we illustrate the wide range of practical problems that can be formulated as (PI). These examples are further explored in Section 4 and more details on their formulations are provided in Appendix E.

1. **Sampling from convex sets**: though we have stressed that (PI) accommodates other types of requirements, it can also be used to constrain the support of $\pi$, i.e., to sample from

$$\mu^\star \in \underset{\mu \in \mathcal{P}_2(\mathbb{R}^d)}{\arg\min} \quad \mathrm{KL}(\mu\|\pi)$$
$$\text{subject to} \quad \mathbb{P}_{x \sim \mu}[x \in \mathcal{C}] = 1, \tag{PII}$$

   for a closed convex set $\mathcal{C} \subset \mathbb{R}^d$. Indeed, let $\mathcal{C}$ be the intersection of the 0-sublevel sets of convex functions $\{s_i\}_{i=1,\dots,I}$. Such a description always exists (see Appendix E). Then, (PII) can be cast as (PI) using $g_i(x) = [s_i(x)]_+$ for $[z]_+ = \max(0, z)$. Notice that the $g_i$ are convex and that although they are not everywhere differentiable, $\mathbb{I}(s_i(x) > 0)\nabla s_i(x)$ is a *subgradient* of $g_i$, where $\mathbb{I}(\mathcal{E}) = 1$ on the event $\mathcal{E}$ and 0 otherwise. Observe that support constraints can also be imposed using projections, mirror/proximal maps, and barriers as in [12–18]. These methods, however, constrain the samples $x$ rather than their distribution $\mu$ as in (PII).

2. **Rate-constrained Bayesian models**: rate constraints have garnered attention in ML due to their central role in fairness [20, 21]. Consider data pairs $(x, y)$, where $x \in \mathcal{X}$ are features and $y \in \{0, 1\}$ labels, and a protected (measurable) subgroup $\mathcal{G} \subset \mathcal{X}$. Let $\pi$ be a Bayesian posterior of the parameters $\theta$ of a model $q(\cdot; \theta)$ denoting the probability of a positive outcome (based, e.g., on a binomial model). We wish to enforce statistical parity, i.e., we wish the prevalence of positive outcomes within the protected group $\mathcal{G}$ to be close to or higher than in the whole population. We cast this problem as (PI) by constraining the average probability of positive outcome as in

$$P^\star = \underset{\mu \in \mathcal{P}_2(\mathbb{R}^d)}{\min} \quad \mathrm{KL}(\mu\|\pi)$$
$$\text{subject to} \quad \mathbb{E}_{x,\theta \sim \mu}\big[q(x;\theta) \mid \mathcal{G}\big] \geq \mathbb{E}_{x,\theta \sim \mu}\big[q(x;\theta)\big] - \delta. \tag{PIII}$$

   where $\delta > 0$ denotes our tolerance [22]. Naturally, multiple protected groups can be accommodated by incorporating additional constraints. Hence, constrained sampling provides a natural way to encode fairness in Bayesian inference.

3. **Counterfactual sampling**: rather than imposing requirements on probabilistic models, constrained sampling can also be used to probe them by evaluating *counterfactual* statements. Indeed, let $\pi$ denote a reference probabilistic model such that sampling from $\pi$ yields realizations of the "real world." Consider the *counterfactual* statement "how would the world have been if $\mathbb{E}[g(x)] \leq 0$?" Constrained sampling not only gives realizations of this alternative world, but it also indicates its "compatibility" with the reference model, namely the value $P^\star$ of (PI).

More concretely, consider a *Bayesian stock market* model. Here, $\pi$ is a posterior model for the (log-)returns of $I$ assets, e.g., distributed as Gaussians $\mathcal{N}(\rho, \Sigma)$. Here, the vector $\rho$ describes the mean return of each stock and $\Sigma$ their covariance. We can investigate what the market would look like if, e.g., the mean and variance of each stocks were to change by solving

$$
\begin{aligned}
P^\star = \min_{\mu \in \mathcal{P}_2(\mathbb{R}^d)} \quad & \mathrm{KL}(\mu \| \pi) \\
\text{subject to} \quad & \mathbb{E}_{(\rho, \Sigma) \sim \mu}[\rho_i] = \bar{\rho}_i, \quad i = 1, \ldots, I \\
& \mathbb{E}_{(\rho, \Sigma) \sim \mu}[\Sigma_{ii}] \leq \bar{\sigma}_i^2
\end{aligned} \tag{PIV}
$$

Due to correlations in the market, certain choices of $\bar{\rho}_i$ or $\bar{\sigma}_i^2$ may be more "unrealistic" than others. Additionally, it could be that some of these conditions are vacuous conditioned on the others. As we show next, our approach to tackling (PI) effectively isolates the contribution of each requirement in the solution $\mu^\star$, thus enabling us to identify which are (conditionally) vacuous and which are most at odds with the reference model $\pi$.

## 2.3 Lagrangian duality and dual ascent algorithms

Although directly sampling from $\mu^\star$ does not appear straightforward, it admits a convenient characterization based on convex duality that is amenable to be sampled using the LMC algorithm (3). Indeed, let $g : \mathbb{R}^d \to \mathbb{R}^I$ and $h : \mathbb{R}^d \to \mathbb{R}^J$ be vector-valued functions collecting the constraint functions $g_i$ and $h_j$ respectively. The Lagrangian of (PI) is then defined as

$$
L(\mu, \lambda, \nu) \triangleq \mathrm{KL}(\mu \| \pi) + \lambda^\top \mathbb{E}_\mu[g] + \nu^\top \mathbb{E}_\mu[h] = \mathrm{KL}(\mu \| \mu_{\lambda\nu}) + \log\left(\frac{Z}{Z_{\lambda\nu}}\right), \tag{4}
$$

for $\lambda \in \mathbb{R}_+^I$ and $\nu \in \mathbb{R}^J$, where

$$
\mu_{\lambda\nu}(x) = \frac{e^{-U(x, \lambda, \nu)}}{Z_{\lambda\nu}} \quad \text{for} \quad U(x, \lambda, \nu) = f(x) + \lambda^\top g(x) + \nu^\top h(x) \tag{5}
$$

and a normalization constant $Z_{\lambda\nu}$. Notice that $P^\star = \min_\mu \max_{\lambda \geq 0, \nu} L(\mu, \lambda, \nu)$, which is why (PI) is referred to as the *primal problem*.

To obtain the *dual problem* of (PI), define the dual function

$$
d(\lambda, \nu) \triangleq \min_{\mu \in \mathcal{P}_2(\mathbb{R}^d)} L(\mu, \lambda, \nu). \tag{6}
$$

Notice from (4) that the minimum in (6) is achieved for $\mu_{\lambda\nu}$ from (5), the *Lagrangian minimizer*, so that $d(\lambda, \nu) = \log(Z/Z_{\lambda\nu})$. The solution of (6) is therefore a *tilted* version of $\pi$, whose tilt is controlled by the *dual variables* $(\lambda, \nu)$. Since (6) is a relaxation of (PI), it yields a lower bound on the primal value, i.e., $d(\lambda, \nu) \leq P^\star$ for all $(\lambda, \nu) \in \mathbb{R}_+^I \times \mathbb{R}^J$. The dual problem seeks the tilts $(\lambda^\star, \nu^\star)$ that yield the best lower bound, i.e.,

$$
D^\star \triangleq \max_{\lambda \in \mathbb{R}_+^I, \nu \in \mathbb{R}^J} d(\lambda, \nu). \tag{DI}
$$

The set $\Phi^\star = \mathrm{argmax}_{\lambda \geq 0, \nu} d(\lambda, \nu)$ of solutions of (DI) is called the set of *Lagrange multipliers*. Note from (6) that (DI) depends on the distributions $\mu$ and $\pi$ through its objective $d$.

The dual problem (DI) has several advantageous properties. Indeed, while the primal problem (PI) is an *infinite dimensional, smooth* optimization problem in probability space, the dual problem (DI) is a *finite dimensional, non-smooth* optimization problem in Euclidean space. What is more, it is a concave problem regardless of the functions $f, g, h$, since the dual function (6) is the minimum of a set of affine functions in $(\lambda, \nu)$ [30, Prop. 4.1.1]. These properties are all the more attractive given that, under mild conditions stated below, (DI) can be used to solve (PI).

**Assumption 2.1.** There exists $\mu^\dagger \in \mathcal{P}_2(\mathbb{R}^d)$ with $\mathrm{KL}(\mu^\dagger \| \pi) \leq C < \infty$ such that $\mathbb{E}_{\mu^\dagger}[g_i] \leq -\delta < 0$ and $\mathbb{E}_{\mu^\dagger}[h_j] = 0$ for all $i, j$.

**Proposition 2.2.** *Under Assumption 2.1, the following holds:*

(i) $P^\star = D^\star$;

(ii) *there exists a finite pair* $(\lambda^\star, \nu^\star) \in \Phi^\star$;

(iii) *for any solution* $\mu^\star$ *of* (PI) *and* $(\lambda^\star, \nu^\star)$ *of* (DI)*, it holds that*

$$L(\mu^\star, \lambda, \nu) \leq L(\mu^\star, \lambda^\star, \nu^\star) \leq L(\mu, \lambda^\star, \nu^\star), \quad \text{for all } (\mu, \lambda, \nu) \in \mathcal{P}_2(\mathbb{R}^d) \times \mathbb{R}_+^I \times \mathbb{R}^J; \quad (7)$$

(iv) *the solution of* (PI) *is* $\mu^\star = \mu_{\lambda^\star \nu^\star}$ *for* $(\lambda^\star, \nu^\star) \in \Phi^\star$;

(v) *consider the perturbation of* (PI)

$$P^\star(u, v) \triangleq \min_{\mu \in \mathcal{P}_2(\mathbb{R}^d)} \mathrm{KL}(\mu \| \pi) \text{ subject to } \mathbb{E}_{x \sim \mu}[g_i(x)] \leq u_i, \ \mathbb{E}_{x \sim \mu}[h_j(x)] = v_j. \quad \text{(PV)}$$

*Then,* $(\lambda^\star, \nu^\star)$ *are subgradients of* $P^\star(0, 0) = P^\star$, *i.e.,* $P^\star(u, v) \geq P^\star - \lambda^{\star\top} u - \nu^{\star\top} v$, *and if* $P^\star(u, v)$ *is differentiable at* $(0, 0)$*, then* $\nabla_u P^\star(u, v) = -\lambda^\star$ *and* $\nabla_v P^\star(u, v) = -\nu^\star$ *at* $(0, 0)$.

*Proof.* In finite dimensional settings, (i)–(v) are well-known duality results (see, e.g., [30]). While they also hold for infinite dimensional optimization problems, their proofs are slightly more "scattered." We collect their reference below. The objective of (PI) is a convex function and its constraints are linear functions of $\mu$. Hence, (PI) is a convex program. Under Slater's condition (Assumption 2.1), it is (i) strongly dual ($P^\star = D^\star$) and (ii) there exists at least one solution $(\lambda^\star, \nu^\star)$ of (DI) (see [31, Sec. 8.6, Thm. 1] or [32, Cor. 4.1]). This implies (iii) the existence of the saddle-point (7) [33, Prop. 2.156], (iv) that $\mu^\star \in \mathrm{argmin}_\mu L(\mu, \lambda^\star, \nu^\star) = \{\mu_{\lambda^\star \nu^\star}\}$, since the KL divergence is strongly convex and its minimizer is unique [34, Thm. 7.3.7], and (v) that $(\lambda^\star, \nu^\star)$ are subgradients of the perturbation function $P^\star(u, v)$ [33, Prop. 4.27]. ∎

Proposition 2.2 shows that given solutions $(\lambda^\star, \nu^\star)$ of (DI), the constrained sampling problem (PI) reduces to sampling from $\mu_{\lambda^\star \nu^\star} \propto e^{-U(\cdot, \lambda^\star, \nu^\star)}$ (see Appendix F for an explicit example of this result). It is important to note that this results only relies on the KL divergence being (strongly) convex in the standard $L^2$ geometry, i.e., along mixtures of the form $t\mu_0 + (1 - t)\mu_1$ for $t \in [0, 1]$. This does not imply that it is (geodesically) convex in the Wasserstein sense [35, Section 9.1.2]. This would require $U$ in (5) to be convex for all $\lambda \geq 0$ and $\nu \in \mathbb{R}^J$, i.e., for $f, g$ to be convex and $h$ to be linear.

Hence, Proposition 2.2 reduces the constrained sampling problem (PI) to that of finding the Lagrange multipliers $(\lambda^\star, \nu^\star)$. Despite their finite dimensionality, however, computing these parameters is intricate. Indeed, since (DI) is a concave program, we could obtain $(\lambda^\star, \nu^\star) \in \Phi^\star$ using

$$\lambda_{k+1} = \left[\lambda_k + \eta_k \mathbb{E}_{\mu_{\lambda_k \nu_k}}[g]\right]_+ \quad \text{and} \quad \nu_{k+1} = \nu_k + \eta_k \mathbb{E}_{\mu_{\lambda_k \nu_k}}[h], \quad (8)$$

for $\eta_k > 0$, where we used the fact that $\mathbb{E}_{\mu_{\lambda \nu}}[g]$ and $\mathbb{E}_{\mu_{\lambda \nu}}[h]$ are (sub)gradients of the dual function (6) at $(\lambda, \nu)$ [36, Thm. 2.87]. This procedure is known in optimization and game theory as *dual ascent* or *best response* [37, 38]. Notice, however, that (8) is not a practical algorithm as it requires explicit integration with respect to the intractable distribution $\mu_{\lambda \nu}$ from (5).

This issue was partially addressed in [24] (in continuous time and without equality constraints, i.e., $J = 0$) by replacing the Lagrangian minimizer $\mu_{\lambda_k \nu_k}$ in (8) by the distribution of LMC samples, as in

$$\begin{aligned}
x_{k+1} &= x_k - \gamma_k \nabla U(x_k, \lambda_k) + \sqrt{2\gamma_k} \beta_k, \quad \beta_k \overset{\text{iid}}{\sim} \mathcal{N}(0, \mathrm{I}_d) \\
\lambda_{k+1} &= \left[\lambda_k + \eta_k \mathbb{E}_{\mu_k}[g]\right]_+, \quad \mu_k = \mathrm{Law}(x_k).
\end{aligned} \quad (9)$$

Note that since $J = 0$, we omit the argument $\nu$ of $U$ for clarity. Nevertheless, the updates in (9) still require an explicit integration. While it is now possible to sample from $\mu_k$ (namely, using the $x_k$), empirical approximations of $\mathbb{E}_{\mu_k}[g]$ may not only require an exponential (in the dimension $d$) number of samples (e.g., [39, Thm. 1.2]), but it introduces errors that are not taken into account in the analysis of [24]. In the sequel, we address these drawbacks by replacing these dual ascent algorithms by a saddle-point one.

---

**Algorithm 1** Primal-dual LMC

---

1: **Inputs:** $\eta_k > 0$ (step size), $x_0 \sim \mu_0$, and $(\lambda_0, \nu_0) = (0, 0)$.
2: **for** $k = 0, \ldots, K - 1$
3:     $x_{k+1} = x_k - \eta_k \nabla_x U(x_k, \lambda_k, \nu_k) + \sqrt{2\eta_k}\, \beta_k, \quad$ for $\beta_k \sim \mathcal{N}(0, \mathrm{I}_d)$
4:     $\lambda_{k+1} = \left[ \lambda_k + \eta_k g(x_k) \right]_+$
5:     $\nu_{k+1} = \nu_k + \eta_k h(x_k)$
6: **end**

---

## 3  Primal-dual Langevin Monte Carlo

Consider the GDA dynamics for the saddle-point problem (DI) in Wasserstein space. Explicitly,

$$\frac{\partial \mu(t)}{\partial t} = \nabla \cdot \left[ \mu(t) \nabla_{W_2} L\big(\mu(t), \lambda(t), \nu(t)\big) \right] \tag{10a}$$

$$\frac{\partial \lambda_i(t)}{\partial t} = \left[ \nabla_{\lambda_i} L\big(\mu(t), \lambda(t), \nu(t)\big) \right]_{\lambda_i(t),+} \tag{10b}$$

$$\frac{\partial \nu_j(t)}{\partial t} = \nabla_{\nu_j} L\big(\mu(t), \lambda(t), \nu(t)\big) \tag{10c}$$

for the Lagrangian $L$ defined in (4), where $[z]_{\lambda,+} = z$ for $\lambda > 0$ and $[z]_{\lambda,+} = \max(a, 0)$ otherwise [40, Sec. 2.2]. Observe that $\nabla_{\lambda_i} L(\mu, \lambda, \nu) = \mathbb{E}_\mu[g_i]$ and $\nabla_{\nu_j} L(\mu, \lambda, \nu) = \mathbb{E}_\mu[h_j]$. Hence, the algorithm from [24] described in (9) involves a *deterministic* implementation of (10b) that fully integrates over $\mu(t)$. In contrast, we consider a *stochastic*, *single-particle* implementation of (10) that leads to the practical procedure in Algorithm 1.

Explicitly, we also use an Euler-Maruyama time-discretization of the Langevin diffusion associated to (10a) (step 3), but replace the expectations in (10b)–(10c) by single-sample approximations (steps 4–5). Algorithm 1 can therefore be seen as a particle implementation of the deterministic Wasserstein GDA algorithm (10). As such, it resembles a primal-dual counterpart of the LMC algorithm in (3), which is why we dub it *primal-dual LMC* (PD-LMC). Alternatively, Algorithm 1 can be interpreted as a stochastic approximation of the dual ascent method in (9). This suggests that the gradient approximations in steps 4–5 could be improved using mini-batches, which is in fact how [24] approximates the expectation in (9). Our theoretical analysis and experiments show that these mini-batches are neither necessary nor always worth the additional computational cost (see Section 3.1 and Section 4). Note that the "stochastic approximations" in Algorithm 1 refer to the dual updates (steps 4–5) rather than the LMC update (step 3) as in stochastic gradient Langevin [41]. Though these methods could be combined, it is beyond the scope of this work.

The remainder of this section is dedicated to analyzing the convergence properties of PD-LMC for both stochastic dual gradients (as in Algorithm 1) and exact dual gradient (as in (9)). For the latter, we obtain guarantees for the discrete implementation (9) under weaker assumptions than the continuous-time analysis of [24]. We consider strongly log-concave target distributions in Section 3.1 and those satisfying an LSI in Section 3.2.

### 3.1  PD-LMC with (strongly) convex potentials

As opposed to the traditional LMC algorithm (3) or the deterministic updates in (9), Algorithm 1 involves three coupled random variables, namely, $(x_k, \lambda_k, \nu_k)$. Hence, the LMC update (step 3) is based on a *stochastic* potential $U$ and the distribution $\mu_k$ of $x_k$ is now a *random measure*. Our analysis sidesteps this obstacle by using techniques from stochastic optimization. We also leverage techniques from primal-dual algorithms in the Wasserstein space, in the spirit of works such as [7, 8, 42] that studied the LMC (3) or alternative time-discretizations of gradient flows of the KL divergence as splitting schemes.

First, define the potential energy $\mathcal{E}$ and the (negative) entropy $\mathcal{H}$ for $(\mu, \lambda, \nu) \in \mathcal{P}_2(\mathbb{R}^d) \times \mathbb{R}_+^I \times \mathbb{R}^J$ as

$$\mathcal{E}(\mu, \lambda, \nu) = \int U(x, \lambda, \nu) d\mu(x) \quad \text{and} \quad \mathcal{H}(\mu) = \int \log(\mu) d\mu, \tag{11}$$

for $U$ as in (5). Notice from (4) that $L(\mu, \lambda, \nu) = \mathcal{E}(\mu, \lambda, \nu) + \mathcal{H}(\mu) - \log(Z)$, where we used the KL divergence decomposition in (1). To proceed, consider the following assumptions:

**Assumption 3.1.** The potential energy $\mathcal{E}(\mu, \lambda, \nu)$ in (11) is $m$-strongly convex with respect to $\mu$ along Wasserstein-2 geodesics for $m \geq 0$ and all $(\lambda, \nu) \in \mathbb{R}_+^I \times \mathbb{R}^J$. Explicitly,

$$\mathcal{E}(\mu, \lambda, \nu) \geq \mathcal{E}(\mu_0, \lambda, \nu) + \int \langle \nabla_{W_2} \mathcal{E}(\mu_0, \lambda, \nu), x - y \rangle ds(x, y) + \frac{m}{2} W_2^2(\mu, \mu_0),$$

where $s$ is an optimal coupling achieving $W_2^2(\mu, \mu_0)$ (see Appendix B).

**Assumption 3.2.** The gradients and variances of $f, g, h$ are bounded along iterations $\{\mu_k\}_{k \geq 0}$, where $\mu_k$ is the distribution of $x_k$, i.e., there exists $G^2$ such that

$$\max\left( \|\nabla f\|_{L^2(\mu_k)}^2, \|\nabla g_i\|_{L^2(\mu_k)}^2, \|\nabla h_j\|_{L^2(\mu_k)}^2 \right) \leq G^2 \quad \text{and} \quad \max\left( \mathbb{E}_{\mu_k}[\|g\|^2], \mathbb{E}_{\mu_k}[\|h\|^2] \right) \leq G^2.$$

Assumption 3.1 holds with $m = 0$ if $f, g$ are convex and $h$ is linear. If $f$ is additionally strongly convex, then it holds with $m > 0$ [26, Prop. 9.3.2]. Assumption 3.2 is typical in (stochastic) non-smooth optimization analyses (see, e.g., [36, 43, 44]). Notice, however, that gradients are only required to be bounded along trajectories of Algorithm 1, a crucial distinction in the case of strongly convex functions whose gradients can only be bounded locally. Assumption 3.2 can be satisfied under mild conditions on $f, g, h$, such as local Lipschitz continuity or linear growth.

The following theorem provides the first convergence analysis of the discrete-time PD-LMC.

**Theorem 3.3.** *Denote by $\mu_k$ the distribution of $x_k$ in Algorithm 1. Under Assumptions 2.1, 3.1, and 3.2, there exists $R_0^2$ such that, for $\eta_k \leq \eta$,*

$$\frac{1}{K} \sum_{k=1}^{K} \left[ \mathrm{KL}(\mu_k \| \mu^\star) + \frac{m}{2} W_2^2(\mu_k, \mu^\star) \right] \leq 3\eta G^2 + \frac{R_0^2}{\eta K} + \frac{\eta G^2}{K} \sum_{k=1}^{K} \left( \mathbb{E}[\|\lambda_k\|^2] + \mathbb{E}[\|\nu_k\|^2] \right). \quad (12)$$

*For $\eta_k \leq R_0/(G\sqrt{k})$ and $\bar{\eta}_k = \eta_k / \sum_{k=1}^{K} \eta_k$, we obtain*

$$\sum_{k=1}^{K} \bar{\eta}_k \left[ \mathrm{KL}(\mu_k \| \mu^\star) + \frac{m}{2} W_2^2(\mu_k, \mu^\star) \right] \leq \frac{R_0 G(1 + \log(K))}{\sqrt{K}} \left( 3 + \max_k \left\{ \mathbb{E}[\|\lambda_k\|^2] + \mathbb{E}[\|\nu_k\|^2] \right\} \right).$$

$$(13)$$

*Additionally, there exists a sequence of step sizes $\eta_k > 0$ such that $W_2^2(\mu_k, \mu^\star) \leq R_0^2$ and $\mathbb{E}[\|\lambda_k\|^2] + \mathbb{E}[\|\nu_k\|^2] < \infty$ for all $k$. The same results hold (without expectations) when using exact dual gradients, i.e., if the updates in steps 4–5 are replaced by $\lambda_{k+1} = [\lambda_k + \eta_k \mathbb{E}_{\mu_k}[g]]_+$ and $\nu_{k+1} = \nu_k + \eta_k \mathbb{E}_{\mu_k}[h]$.*

Theorem 3.3, whose proof is deferred to Appendix C, implies rates similar to those for GDA schemes in finite-dimensional Euclidean optimization (see, e.g., [43]). To recover those rates, however, we must bound the magnitudes of $\lambda_k, \nu_k$. In [43], this is done by bounding the iterates in the algorithm itself, i.e., by projecting them onto the set $\mathcal{D}_r = \{(\lambda, \nu) \in \mathbb{R}_+^I \times \mathbb{R}^J \mid \max(\|\lambda\|^2, \|\nu\|^2) \leq r\}$ and choosing $r$ such that $\Phi^\star \subseteq \mathcal{D}_r$ (Proposition 2.2(ii) ensures this is possible). We then incur a bias on the order of $\eta$ in (12) that vanishes in the decreasing step size setting of (13). Though convenient, this is not *necessary* since there exists a sequence of step sizes such that both $\mathbb{E}[\|\lambda_k\|^2]$ and $\mathbb{E}[\|\nu_k\|^2]$ are bounded for all $k \geq 0$. In the interest of generality, Theorem 3.3 holds without these hypotheses. It is worth noting that though faster rates and last iterates guarantees can be obtained for Euclidean saddle-point problems, they rely on more complex schemes than the GDA in Algorithm 1 involving acceleration or proximal methods [45–48].

The results in Theorem 3.3 are stated for the stochastic scheme in Algorithm 1. However, Theorem 3.3 yields the same rates (without expectations) for exact dual gradients, i.e., for the dual ascent scheme (9). In this case, the second condition in Assumption 3.2 simplifies to $\max_k(\|\mathbb{E}_{\mu_k}[g]\|^2, \|\mathbb{E}_{\mu_k}[h]\|^2) \leq G^2$. Not only are these milder assumptions than [24, Eq. (16)], but the guarantees hold for discrete- rather than continuous-time dynamics. Finally, (12)–(13) imply convergence with respect to the KL divergence for convex potentials ($m = 0$) with stronger guarantees in Wasserstein metric for strongly convex ones ($m > 0$).

The convergence rates for distributions $\mu_k$ from Theorem 3.3 also imply convergence rates for empirical averages across iterates $x_k$ of Algorithm 1. This corollary is obtained by combining (12)–(13) with the following proposition. By taking $\varphi$ to be the constraint functions $g$ or $h$ from (PI) yields feasibility guarantees for PD-LMC .

---

**Algorithm 2** (Stochastic) dual LMC

---

1: **Inputs:** $N_k^0 > 0$ (burn-in), $\gamma_k, \eta_k > 0$ (step sizes), and $\lambda_0 = 0$.
2: **for** $k = 0, \ldots, K - 1$
3:     $x_0 \sim \mu_0$
4:     $x_{n+1} = x_n - \gamma_k \nabla_x U(x_n, \lambda_k) + \sqrt{2\gamma_k}\, \beta_n, \quad \beta_n \sim \mathcal{N}(0, \mathrm{I}_d), \quad$ for $n = 0, \ldots, N_k^0 - 1$
5:     $\lambda_{k+1} = \left[ \lambda_k + \eta_k g(x_{N_k^0}) \right]_+$
6: **end**

---

**Proposition 3.4.** *Consider samples $x_k$ distributed according to $\mu_k$ and $c_k \geq 0$ with $\sum_{k=1}^{K} c_k = 1$ such that $\sum_{k=1}^{K} c_k \left[ \mathrm{KL}(\mu_k \,\|\, \mu^\star) + \frac{m}{2} W_2^2(\mu_k, \mu^\star) \right] \leq \Delta_K$ for $m \geq 0$. Then, it holds that*

$$\left| \mathbb{E}\left[ \sum_{k=1}^{K} c_k \varphi(x_k) \right] - \mathbb{E}_{\mu^\star}[\varphi] \right| \leq \begin{cases} \sqrt{2\Delta_K}, & \text{if } \varphi \text{ is bounded by 1}, \\ \sqrt{\dfrac{2\Delta_K}{m}}, & \text{if } \varphi \text{ is 1-Lipschitz and } m > 0. \end{cases}$$

*Proof.* See Appendix C.

### 3.2 PD-LMC with LSI potentials

In this section, we replace Assumption 3.1 on the convexity of the potential by an LSI common in the sampling literature. We consider only inequality constraints ($J = 0$) here and omit the function arguments $\nu$, since accounting for equality constraints requires significant additional assumptions.

**Assumption 3.5.** *The distribution $\mu_\lambda$ satisfies the LSI for bounded $\lambda$, i.e., there exists $\sigma > 0$ such that $2\sigma \,\mathrm{KL}(\zeta \| \mu_\lambda) \leq \|\nabla \log (d\zeta/d\mu_\lambda)\|_{L^2(\zeta)}^2$ for all $\zeta \in \mathcal{P}_2(\mathbb{R}^d)$.*

The LSI in Assumption 3.5 is often used in the analysis of the standard LMC algorithm [29, 49]. It holds, e.g., when $f$ is strongly convex and $g$ is a (possibly non-convex) bounded function due to the Holley-Stroock perturbation theorem [50]. In fact, if $f$ is 1-strongly convex and $|g|$ is bounded by 1, then Assumption 3.5 holds for $\sigma \geq e^{-2\lambda}$ (see, e.g., [51, Prop. 5.1.6] or [52, Thm 1.1]). The LSI is akin to the Polyak-Łojasiewicz (PL) condition from Euclidean optimization [53], which supposes issues with GDA methods such as Algorithm 1. Indeed, it is not enough for the Lagrangian (4) to satisfy the PL condition in the primal variable to guarantee the convergence of GDA in Euclidean spaces. We must either modify Algorithm 1 using acceleration or proximal methods [47, 54–56] or impose the PL condition also on $\lambda$ [57, 58]. Since the Lagrangian 4 is linear in $\lambda$, it is clear that Algorithm 1 will not suffice to provide theoretical guarantees in the LSI case.

We therefore consider the variant in Algorithm 2, where $N_k^0$ LMC iterations (step 3) are executed before updating the dual variables (step 4). This is akin to using different time-scales in continuous-time, a common technique for solving saddle-point problems [54, 58]. Since it resembles a dual ascent counterpart of the LMC algorithm (3), we refer to it as *(stochastic) dual LMC* (DLMC). As opposed to the dual ascent algorithm from [24] in (9), however, Algorithm 2 does not require any explicit evaluation of expected values. The following theorem provides an analysis of its convergence.

**Theorem 3.6.** *Assume that the functions $f, g$ are $M$-smooth, i.e., have $M$-Lipschitz continuous gradients, satisfy Assumption 3.5, and that $\mathbb{E}_\mu[\|g\|^2] \leq G^2$ for all $\mu \in \mathcal{P}_2(\mathbb{R}^d)$. Let $0 < \eta_k \leq \eta$, $0 < \epsilon \leq \eta G^2 < 1$,*

$$\gamma_k = \gamma \leq \frac{\sigma\epsilon}{16dM^2}, \quad \text{and} \quad N_k^0 \geq \frac{1}{\gamma\sigma} \log\left( \frac{2\,\mathrm{KL}(\mu_0 \| \mu_{\lambda_k})}{\epsilon} \right).$$

*Under Assumption 2.1, there exists $B < \infty$ such that the distributions $\{\mu_k\}$ of the samples $\{x_{N_k^0}\}$ generated by Algorithm 2 satisfy*

$$\frac{1}{K} \sum_{k=0}^{K-1} \mathrm{KL}(\mu_k \| \mu^\star) \leq \epsilon + \frac{\eta G^2}{2} + \frac{2IB^2}{\eta K}. \tag{14}$$

*Recall from* (PI) *that $I$ is the number of inequality constraints. Additionally, $\mathbb{E}[\|\lambda_k\|_1]$ is bounded for all $k$.*

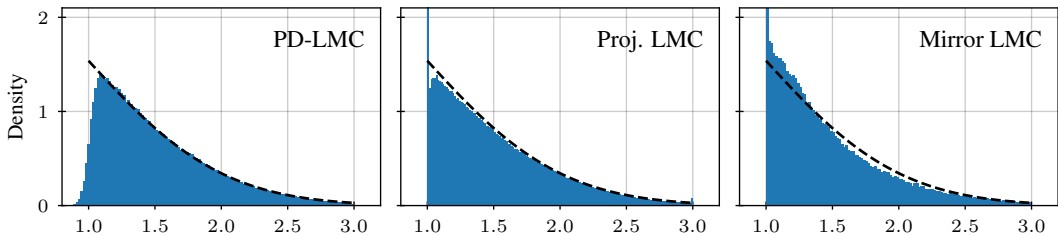

Figure 1: Sampling from a 1D truncated Gaussian (ground truth displayed as dashed lines).

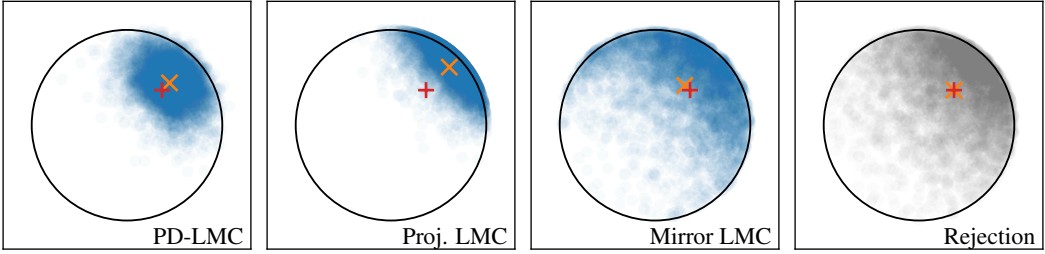

Figure 2: Sampling from a 2D truncated Gaussian (true mean in red and sample mean in orange).

Theorem 3.6, whose proof is deferred to Appendix D, provides similar guarantees as (approximate) subgradient methods in finite-dimensional optimization (see, e.g., [22, 56]). This is not surprising seen as $\gamma_k, N_k^0$ in Theorem 3.6 are chosen to ensure that step 4 yields a sample $x_k \sim \bar{\mu}_k$ such that $\mathrm{KL}(\bar{\mu}_k \| \mu_{\lambda_k}) \leq \epsilon$ using [29, Theorem 1]. At this point, $g(x_{N_k^0})$ in step 5 is an approximate, stochastic subgradient of the dual function (6). Though it may appear from (12) and (14) that Algorithms 1 and 2 have the same convergence rates, an informal computation shows that the latter evaluates on the order of $d\kappa^2/\eta$ as many gradient per iteration, where $\kappa = M/\sigma$. Note that we can once again apply Theorem 3.6 to derive ergodic average and feasibility guarantees for Algorithm 2.

## 4    Experiments

We now return to the applications described in Section 2.2 to showcase the behavior of PD-LMC. We defer implementation details and additional results to Appendix E. Code for these examples is publicly available at `https://www.github.com/lfochamon/pdlmc`.

**1. Sampling from convex sets.** We cast the problem of sampling from a Gaussian distribution $\mathcal{N}(0,1)$ truncated to $\mathcal{C} = [1,3]$ as (PI) by taking $f(x) = x^2/2$ and $g(x) = [(x-1)(x-3)]_+$ (see Section 2.2). Fig. 1 shows histograms for the samples obtained using PD-LMC, the projected LMC (Proj. LMC) from [13], and the mirror LMC from [59], all with the same step size. Both Proj. LMC and Mirror LMC generate an excess of samples close to the boundary (between 1.5 and 3 times more samples than expected). This leads to an underestimation of the mean (Proj. LMC: 1.488 / Mirror LMC: 1.470 vs. true mean: 1.510). In contrast, PD-LMC provides a more accurate estimate (1.508). Yet, since it constrains the distribution $\mu$ rather than its samples, it is not an *interior-point method* and can produce samples outside of $\mathcal{C}$. Theorems 3.3–3.6 show that this becomes less frequent as the algorithm progresses (in Fig. 1, only 2% of the samples are not in $\mathcal{C}$). This occurs even without using *mini-batches* in steps 4–5 of Algorithm 1 as in [24]. In fact, our experiments show that *mini-batches* increase the computational complexity with no performance benefit (Appendix E). These issues are exacerbated in more challenging problems, such as sampling from a two-dimensional standard Gaussian centered at $[2,2]$ restricted to an unit $\ell_2$-norm ball (Fig. 2). In this case, Proj. LMC places almost 25% of its samples on the boundary (where only 0.14% of samples should be), while PD-LMC only places 1.8% of its samples outside of the support. Mirror LMC provides a better mean estimation in this setting, although a bit more asymmetric than PD-LMC [Mirror LMC: $(0.312, 0.418)$ vs. PD-LMC: $(0.446, 0.444)$ vs. true mean: $(0.368, 0.368)$].

**2. Rate-constrained Bayesian models.** Here, we consider $\pi$ to be the posterior of a Bayesian logistic regression model for the Adult dataset from [60], where the goal is to predict whether an individual

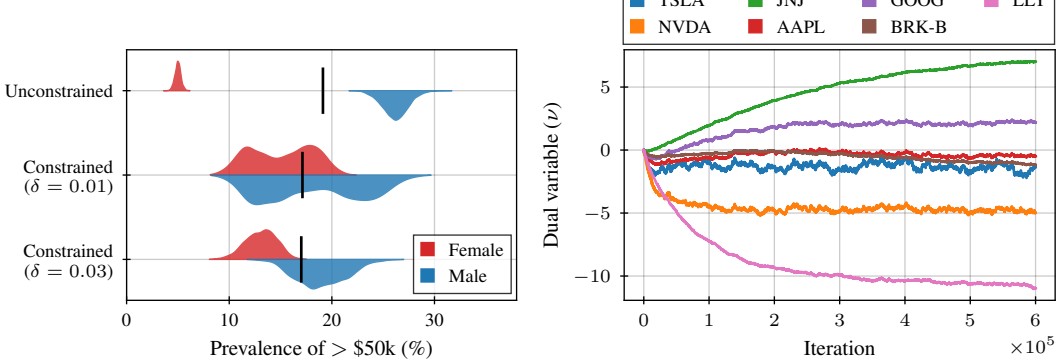

Figure 3: Distribution of the probability of predicting > 50k under the Bayesian logistic model (black lines indicate the mean across genders).

Figure 4: Counterfactual sampling of the stock market: dual variables.

makes more than \$50k based on socioeconomic information (details on data pre-processing can be found in [61]). We consider a standard Gaussian prior on the parameters $\theta \in \mathbb{R}^{d+1}$ of the model, where $d$ is the number of features. Using the LMC algorithm to sample from the posterior (i.e., no constraints), we find that while the average probability of positive predictions is $19.1\%$ over the whole test set, it is $26.2\%$ among males and $5\%$ among females ("Unconstrained" in Fig. 3). To overcome this disparity, we take *gender* to be the protected class in (PIII), constraining both $\mathcal{G}_{\text{male}}$ and $\mathcal{G}_{\text{female}}$ with $\delta = 0.01$. Using PD-LMC, we obtain a Bayesian model that leads to an average probability of positive outcomes of $18.1\%$ and $15.1\%$ for males and females respectively. In fact, we now observe a substantial overlap of the distributions of positive predictions across genders for the constrained posterior $\mu^\star$ ("Constrained ($\delta = 0.01$)" in Fig. 3). This substantial reduction of prediction disparities comes at only a minor decline in accuracy (unconstrained: $84\%$ vs constrained: $82\%$).

**3. Counterfactual sampling.** Though the distribution of positive predictions changes considerably for both male and female individuals, the final dual variables ($\lambda_{\text{male}} = 0$ and $\lambda_{\text{female}} \approx 160$) show that these changes are due uniquely to the *female* group [as per Prop. 2.2(iv)]. This implies that the reference model $\pi$ is itself compatible with the requirement for the male group, but that reducing the disparity for females requires considerable deviations from it. By examining $\lambda_{\text{female}}$, we conclude *without recalculating* $\mu^\star$ that even small changes in the tolerance $\delta$ for the female constraint would substantially change the distribution of outcomes [Prop. 2.2(v)]. This is confirmed by "Constrained ($\delta = 0.03$)" in Fig. 3. Notice that this is only possible due to the primal-dual nature of PD-LMC. This type of counterfactual analysis is even more beneficial in the presence of multiple requirements. Indeed, let $\pi$ be the posterior of a Bayesian model for the daily (log-)return of a set of assets (see Appendix E for more details). Using (PIV), we consider how the market would look like if the average (log-)return of each asset were to have been (exactly) $20\%$ higher. Inspecting the dual variables (Fig. 4), we notice that this increased market return is essentially driven by two stocks: NVDA and LLY ($\nu < 0$). In fact, the reference model $\pi$ would be consistent with an even higher increase for JNJ and GOOG ($\nu > 0$). We confirm these observations by constraining only NVDA and LLY, which yields essentially the same (log-)return distribution for all assets.

## 5  Conclusion

We tackled the problem of sampling from a target distribution while satisfying a set of statistical constraints. Based on a GDA method in Wasserstein space, we put forward a fully stochastic, discrete-time primal-dual LMC algorithm (PD-LMC) that precludes any explicit integration in its updates. We analyze the behavior of PD-LMC for (strongly) convex and log-Sobolev potentials, proving that the distribution of its samples converges to the optimal constrained distribution. We illustrated the use of PD-LMC for different constrained sampling applications. Future work include strengthening the convergence results to almost sure guarantees and improving the rates obtained using proximal and extra gradient methods, particularly in the LSI setting.

## Acknowledgments and Disclosure of Funding

The work of L.F.O. Chamon is funded by the Deutsche Forschungsgemeinschaft (DFG, German Research Foundation) under Germany's Excellence Strategy (EXC 2075-390740016).

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

# A Related work

In constrained sampling, it is important to distinguish between two types of constraints: support constraints and statistical constraints. The former deals with sampling from a target distribution $\pi$ that is supported on a proper subset $\mathcal{X} \subset \mathbb{R}^d$, which arises in applications such as latent Dirichlet allocation [62] and regularized regression [63]. The latter is the problem tackled in the current work.

A first family of constrained sampling methods relies on rejection sampling: it obtains samples via any (unconstrained) method, rejecting those that violate the constraint (see, e.g., [10, 11]). Though this approach can handle constraints of any nature, it is often inefficient in terms of number of samples generated per iteration of the method (effective number of samples), especially when confronted with intricate constraints and high dimensional problems.

These drawbacks can be addressed for support constraints using techniques inspired by finite-dimensional constrained optimization. Projected LMC, for instance, deals with the problem of sampling from a target distribution restricted to a convex set [13, 64]. Barrier methods have also been used to tackle the same problem [18, 65]. Similarly, mirror and proximal descent versions of popular sampling algorithms such as LMC [12, 14, 15, 66–68] and Stein Variational Gradient Descent (SVGD) [69] have been proposed. Mirror descent algorithms enforce constraints by mapping (mirroring) the primal variables to a space with a different geometry (induced by a Bregman divergence) over which the optimization is carried out. Alternatively, methods adapted to target distributions support on manifolds have also been put forward [18, 70, 71]. In practice, these methods require explicit expressions for the projections, barriers, and mirror maps describing the constraints and are therefore not adapted to handle statistical requirements such as those considered in (PI). Langevin dynamics with constraint violation penalties were considered in [23], although they cannot enforce exact constraint satisfaction.

Statistical (moment) constraints such as those considered in (PI) were investigated in [24]. As we discussed at the end of Section 2.3, this paper considers the combination of LMC and approximate dual ascent shown in (9). It also introduces a similar version of SVGD as well as algorithms based on barriers. Aside from requiring exact integration against intractable measures (namely, $\mu_k$), convergence guarantees for these methods hold under restrictive assumptions on the constraints $g_i$. Additionally, guarantees are derived only for continuous-time (gradient flows) dynamics.

This work is also closely related to saddle-point methods in finite-dimensional optimization. For the general problem of $\max_x \min_y f(x, y)$, the behavior of descent-ascent methods have been investigated under a myriad of scenarios, including for functions $f$ that are (strongly) convex/(strongly) concave [43, 45–48, 72, 73] as well as non-convex/non-concave under, e.g., PL conditions [47, 54, 55, 57, 58]. In general, convergence holds for the ergodic average of iterates [43, 57, 73]. Last-iterate results often require different algorithms, involving proximal point or extra gradient methods and time scale separation [47, 48, 73]. In particular, guarantees for the GDA method used in Algorithm 1 often requires stringent conditions that are hard to enforce for dual problems such as (DI).

# B  Wasserstein space and discrete-time flows

In this subsection we give some background on the Wasserstein spaces and sampling as optimization. For $\mu, \nu \in \mathcal{P}_2(\mathbb{R}^d)$, we define the 2-Wasserstein distance as

$$W_2^2(\mu, \nu) = \inf_{s \in \mathcal{S}(\mu,\nu)} \int \|x - y\|^2 ds(x, y),$$

where $\mathcal{S}(\mu, \nu)$ is the set of couplings between $\mu$ and $\nu$. The metric space $(\mathcal{P}_2(\mathbb{R}^d), W_2)$ is referred to as the *Wasserstein space* [74]. It can be equipped with a Riemannian structure [75]. In this geometric interpretation, the tangent space to $\mathcal{P}_2(\mathbb{R}^d)$ at $\mu$ is included in $L^2(\mu)$ and is equipped with a scalar product defined for $f, g \in L^2(\mu)$ as

$$\langle f, g \rangle_{L^2(\mu)} = \int f(x)g(x)d\mu(x).$$

We use its differential structure as introduced in [75] and [26, Chapter 10]. For the functionals at stake in this paper (e.g., potential energies and negative entropy), the set of subgradients of a functional $\mathcal{F} : \mathcal{P}_2(\mathbb{R}^d) \to \mathbb{R}$ at $\mu \in \mathcal{P}(\mathbb{R}^d)$ is non-empty if $\mu$ satisfies some Sobolev regularity [26, Theorem 10.4.13].

A *Wasserstein gradient flow* of $\mathcal{F}$ is a solution $(\mu(t))_{t \in (0,T)}$, $T > 0$, of the continuity equation

$$\frac{\partial \mu(t)}{\partial t} + \nabla \cdot (\mu(t)v(t)) = 0$$

that holds in the distributional sense, where $v(t)$ is a subgradient of $\mathcal{F}$ at $\mu(t)$. Among the possible processes $(v(t))_t$, one has a minimal $L^2(\mu(t))$ norm and is called the velocity field of $(\mu(t))_t$. In a Riemannian interpretation of the Wasserstein space [75], this minimality condition can be characterized by $v(t)$ belonging to the tangent space to $\mathcal{P}_2(\mathbb{R}^d)$ at $\mu(t)$ denoted $T_{\mu(t)}\mathcal{P}_2(\mathbb{R}^d)$, which is a subset of $L^2(\mu(t))$. The Wasserstein gradient is defined as this unique element, and is denoted $\nabla_{W_2}\mathcal{F}(\mu(t))$. In particular, if $\mu \in \mathcal{P}_2(\mathbb{R}^d)$ is absolutely continuous with respect to the Lebesgue measure, with density in $C^1(\mathbb{R}^d)$ and such that $\mathcal{F}(\mu) < \infty$, $\nabla_{W_2}\mathcal{F}(\mu)(x) = \nabla \mathcal{F}'(\mu)(x)$ for $\mu$-a.e. $x \in \mathbb{R}^d$, where $\mathcal{F}'(\mu)$ denotes the first variation of $\mathcal{F}$ evaluated at $\mu$, i.e. (if it exists) the unique function $\mathcal{F}'(\mu) : \mathbb{R}^d \to \mathbb{R}$ s.t.

$$\lim_{\epsilon \to 0} \frac{1}{\epsilon}(\mathcal{F}(\mu + \epsilon\xi) - \mathcal{F}(\mu)) = \int \mathcal{F}'(\mu)(x)d\xi(x) \tag{15}$$

for all $\xi = \nu - \mu$, where $\nu \in \mathcal{P}_2(\mathbb{R}^d)$.

Now, we denote by $T_{\#}\mu$ the pushforward measure of $\mu \in \mathcal{P}_2(\mathbb{R}^d)$ by the measurable map $T$. We recall that the KL divergence of $\mu$ relative to $\pi$ can be decomposed as (1). The distribution $\mu_k$ of $x_k$ in (3) is known to follow a "forward-flow" splitting scheme [7] of the Fokker-Planck equation in (2), namely

$$\begin{aligned}
\mu_{k+1/2} &= \left[\mathrm{I} - \gamma_k \nabla_{W_2}\mathcal{V}(\mu_k)\right]_{\#}\mu_k \quad &\text{(forward discretization for } \mathcal{V}) \\
\mu_{k+1} &= \mu_{k+1/2} \star \mathcal{N}(0, 2\gamma_k \mathrm{I}_d) \quad &\text{(exact flow for } \mathcal{H}),
\end{aligned} \tag{16}$$

where I denotes the identity map in $L^2(\mu_k)$ and $\nabla_{W_2}\mathcal{V}(\mu_k) = \nabla_x f(x)$.

# C  Proofs from Section 3.1

*Proof of Theorem 3.3.* Consider the potential

$$V(\mu, \lambda, \nu) \triangleq W_2^2(\mu, \mu^\star) + \min_{(\lambda^\star, \nu^\star) \in \Phi^\star} \|\lambda - \lambda^\star\|^2 + \|\nu - \nu^\star\|^2, \tag{17}$$

where $\mu^\star$ is the solution of the constrained sampling problem (PI) and $\Phi^\star$ is the set of solutions of the dual problem (DI). Our goal is to show that $V$ decreases (in some sense) along trajectories of Algorithm 1. We say "in some sense" because contrary to the standard LMC algorithm, the distribution $\mu_k$ of $x_k$ is now a random measure that depends on the random variables $\{\lambda_k, \nu_k\}$. Explicitly, we consider the filtration $\mathcal{F}_k = \sigma(\mu_0, \{\lambda_\ell, \nu_\ell\}_{\ell \leq k})$ and show that $V$ decreases on average when conditioned on $\mathcal{F}_k$. This turns out to be enough to prove Theorem 3.3.

Indeed, notice that $\mu_k \in \mathcal{F}_k$. Hence, the potential energy $\mathcal{E}(\mu_k, \lambda_k, \nu_k) \in \mathcal{F}_k$ and the conditional law $\tilde{\mu}_k = \mathcal{L}(x_k|\mathcal{F}_{k-1})$ evolves as in the regular LMC algorithm (3). That is to say, conditioned on the event $\mathcal{F}_k$, step 5 of Algorithm 1 follows a splitting scheme as (16), i.e.,

$$\tilde{\mu}_{k+1/2} = \left[\mathrm{I} - \eta_k \nabla_{W_2} \mathcal{E}(\mu_k, \lambda_k, \nu_k)\right]_\# \mu_k \tag{18a}$$

$$\tilde{\mu}_{k+1} = \tilde{\mu}_{k+1/2} \star \mathcal{N}(0, 2\gamma \mathrm{I}_d) \tag{18b}$$

Notice that all distributions in (18) are now deterministic. The core of the proof is collected in the following lemma that shows that $V$ is a non-negative supermartingale. Note that Lemma C.1 describes the gap between "half-iterations." This will be inconsequential for our purposes.

**Lemma C.1.** *Under the conditions of Theorem 3.3, we have*

$$\mathbb{E}\left[V(\tilde{\mu}_{k+1/2}, \lambda_{k+1}, \nu_{k+1})|\mathcal{F}_k\right] \leq V(\tilde{\mu}_{k-1/2}, \lambda_k, \nu_k) - 2\eta_k \left[\mathrm{KL}(\tilde{\mu}_k \| \mu^\star) + \frac{m}{2} W_2^2(\tilde{\mu}_k, \mu^\star)\right]$$

$$+ \eta_k^2 \left[\|\nabla_{W_2}\mathcal{E}(\tilde{\mu}_k, \lambda_k, \nu_k)\|_{L^2(\tilde{\mu}_k)}^2 + \mathbb{E}_{x \sim \tilde{\mu}_k}\|g(x)\|^2 + \mathbb{E}_{x \sim \tilde{\mu}_k}\|h(x)\|^2\right]. \tag{19}$$

We defer the proof of Lemma C.1 to show how it implies the bounds in Theorem 3.3. To do so, take the expectation of (19) with respect to $\{\lambda_\ell, \nu_\ell\}_{\ell \leq k}$ to obtain

$$\mathbb{E}[\Delta_k] \triangleq \mathbb{E}\left[V(\tilde{\mu}_{k+1/2}, \lambda_{k+1}, \nu_{k+1}) - V(\tilde{\mu}_{k-1/2}, \lambda_k, \nu_k)\right] \leq$$

$$- 2\eta_k \mathbb{E}\left[\mathrm{KL}(\tilde{\mu}_k \| \mu^\star) + \frac{m}{2} W_2^2(\tilde{\mu}_k, \mu^\star)\right]$$

$$+ \eta_k^2 \left[\|\nabla_{W_2}\mathcal{E}(\tilde{\mu}_k, \lambda_k, \nu_k)\|_{L^2(\mu_k)}^2 + \mathbb{E}_{x \sim \mu_k}\|g(x)\|^2 + \mathbb{E}_{x \sim \mu_k}\|h(x)\|^2\right]. \tag{20}$$

Using the bounds in Assumption 3.2 then yields

$$\mathbb{E}[\Delta_k] \triangleq \mathbb{E}\left[V(\tilde{\mu}_{k+1/2}, \lambda_{k+1}, \nu_{k+1}) - V(\tilde{\mu}_{k-1/2}, \lambda_k, \nu_k)\right] \leq$$

$$- 2\eta_k \mathbb{E}\left[\mathrm{KL}(\tilde{\mu}_k \| \mu^\star) + \frac{m}{2} W_2^2(\tilde{\mu}_k, \mu^\star)\right] + \eta_k^2 \left(3 + \mathbb{E}[\|\lambda_k\|^2] + \mathbb{E}[\|\nu_k\|^2]\right) G^2. \tag{21}$$

Then, summing the LHS of (21) over $k$ and using the fact that $V$ is non-negative yields

$$\sum_{k=1}^K \mathbb{E}[\Delta_k] = \mathbb{E}\left[V(\tilde{\mu}_{K+1/2}, \lambda_{k+1}, \nu_{k+1})\right] - V(\tilde{\mu}_{1/2}, \lambda_1, \nu_1) \geq - \mathbb{E}[V(\tilde{\mu}_{1/2}, \lambda_1, \nu_1)]. \tag{22}$$

Notice that the expectation here is taken only over $\mu_0$ given that $(\lambda_0, \nu_0) = (0, 0)$ are deterministic. To proceed, we use the following proposition, whose proof we defer.

**Lemma C.2.** *Under the hypothesis of Theorem 3.3 it holds that $\mathbb{E}[V(\tilde{\mu}_{1/2}, \lambda_1, \nu_1)] \leq R_0^2$ for*

$$R_0^2 = W_2^2(\mu_0, \mu^\star) + \eta_0^2 \left[\|\nabla f\|_{L^2(\mu_0)}^2 + \mathbb{E}_{\mu_0}[\|g\|^2] + \mathbb{E}_{\mu_0}[\|h\|^2]\right] + \|\lambda^\star\|^2 + \|\nu^\star\|^2.$$

Back in (21), we obtain that

$$\sum_{k=1}^{K} \eta_k \left[ \mathbb{E}[\mathrm{KL}(\tilde{\mu}_k \parallel \mu^\star)] + \frac{m}{2} \mathbb{E}[W_2^2(\tilde{\mu}_k, \mu^\star)] \right] \leq R_0^2 + \sum_{k=1}^{K} \eta_k^2 3G^2 + G^2 \sum_{k=1}^{K} \eta_k^2 \left( \mathbb{E}[\|\lambda_k\|^2] + \mathbb{E}[\|\nu_k\|^2] \right).$$

We conclude by using the convexity of the Wasserstein distance to write

$$\mathbb{E}\left[ W_2^2(\tilde{\mu}_{k+1/2}, \mu^\star) \right] \geq W_2^2(\mathbb{E}[\tilde{\mu}_{k+1/2}], \mu^\star) = W_2^2(\mu_{k+1/2}, \mu^\star).$$

Similarly for the KL divergence. The bounds in Theorem 3.3 are then obtained for the specific choices of $\eta_k$ by noticing that $\sum_{k=1}^{K} 1/\sqrt{k} \geq \sqrt{K}$ and $\sum_{k=1}^{K} 1/k \leq 1 + \log(K)$. Notice that all inequalities in the proof continue to hold for deterministic $\{\lambda_k, \nu_k\}$. The bounds in Theorem 3.3 therefore also hold (without expectations) when using exact gradients to update the dual variables.

Finally, we show there exists a sequence of step sizes $\eta_k$ such that $\mathbb{E}[V(\mu_{k-1/2}, \lambda_k, \nu_k)] \leq R_0^2$ for all $k \geq 1$, where the expectation is taken over the $\{\lambda_k, \nu_k\}$. This immediately implies that $W_2^2(\mu_k, \mu^\star) \leq R_0^2$ and both $\mathbb{E}[\|\lambda_k\|^2]$ and $\mathbb{E}[\|\nu_k\|^2]$ are bounded for all $k$. We proceed by induction. The base case is covered by Lemma C.2. Suppose now that there exists a sequence $\{\eta_0, \ldots, \eta_{k-1}\}$ such that $\mathbb{E}[V(\tilde{\mu}_{k-1/2}, \lambda_k, \nu_k)] \leq R_0^2$. From the definition of $V$ in (17) and the fact that the $(\lambda^\star, \nu^\star)$ are bounded (Prop. 2.2), we then obtain that $\mathbb{E}[\|\lambda_k\|], \mathbb{E}[\|\nu_k\|]$ are bounded. Consequently, there exists $\eta_k > 0$ such that

$$\eta_k^2 (3 + \mathbb{E}[\|\lambda_k\|^2] + \mathbb{E}[\|\nu_k\|^2]) G^2 \leq 2\eta_k \, \mathbb{E}\left[ \mathrm{KL}(\tilde{\mu}_k \parallel \mu^\star) + \frac{m}{2} W_2^2(\tilde{\mu}_k, \mu^\star) \right].$$

From (21), we obtain that $\mathbb{E}[\Delta_k] \leq 0$, which together with the induction assumption yields $\mathbb{E}\left[ V(\tilde{\mu}_{k+1/2}, \lambda_{k+1}, \nu_{k+1}) \right] \leq R_0^2$. ∎

*Proof of Lemma C.1.* The proof proceeds by combining two inequalities bounding the primal and dual terms in (17).

**(i) $W_2^2(\tilde{\mu}_{k+1/2}, \mu^\star)$.** We proceed following a coupling argument. Let $s^k$ be an optimal coupling between the random variables $Y \sim \tilde{\mu}_k$ and $Z \sim \mu^\star$, i.e., a coupling that achieves $W_2^2(\tilde{\mu}_k, \mu^\star)$. Consider now the random variable $T = Y - \eta_k \nabla_{W_2} \mathcal{E}(\tilde{\mu}_k, \lambda_k, \nu_k)$ and observe from (18a) that it is distributed as $\tilde{\mu}_{k+1/2}$. Naturally, the coupling $s^k$ is no longer optimal for $(T, Z)$, so that by the definition of the Wasserstein distance it follows that

$$W_2^2(\tilde{\mu}_{k+1/2}, \mu^\star) \leq \int \|x - \eta_k \nabla_{W_2} \mathcal{E}(\tilde{\mu}_k, \lambda_k, \nu_k) - y\|^2 ds^k(x, y). \tag{23}$$

Expanding the RHS of (23) and using the $m$-strong convexity of $\mathcal{E}$ (Assumption 3.1) yields

$$\begin{aligned} W_2^2(\tilde{\mu}_{k+1/2}, \mu^\star) \leq{} & W_2^2(\tilde{\mu}_k, \mu^\star) - \eta_k m W_2^2(\tilde{\mu}_k, \mu^\star) \\ & + 2\eta_k \left[ \mathcal{E}(\mu^\star, \lambda_k, \nu_k) - \mathcal{E}(\tilde{\mu}_k, \lambda_k, \nu_k) \right] \\ & + \eta_k^2 \|\nabla_{W_2} \mathcal{E}(\tilde{\mu}_k, \lambda_k, \nu_k)\|_{L^2(\tilde{\mu}_k)}^2. \end{aligned}$$

We can then bound the effect of the diffusion step using [8, Lemma 5] as in

$$W_2^2(\tilde{\mu}_k, \mu^\star) - W_2^2(\tilde{\mu}_{k-1/2}, \mu^\star) \leq 2\eta_k \left[ \mathcal{H}(\mu^\star) - \mathcal{H}(\tilde{\mu}_k) \right], \tag{24}$$

which yields

$$\begin{aligned} W_2^2(\tilde{\mu}_{k+1/2}, \mu^\star) \leq{} & W_2^2(\tilde{\mu}_{k-1/2}, \mu^\star) - \eta_k m W_2^2(\tilde{\mu}_k, \mu^\star) \\ & + 2\eta_k \left[ \mathcal{E}(\mu^\star, \lambda_k, \nu_k) + \mathcal{H}(\mu^\star) - \mathcal{E}(\tilde{\mu}_k, \lambda_k, \nu_k) - \mathcal{H}(\tilde{\mu}_k) \right] \\ & + \eta_k^2 \|\nabla_{W_2} \mathcal{E}(\tilde{\mu}_k, \lambda_k, \nu_k)\|_{L^2(\tilde{\mu}_k)}^2. \end{aligned} \tag{25}$$

**(ii) $\|\lambda_{k+1} - \lambda^\star\|^2 + \|\nu_{k+1} - \nu^\star\|^2$.** Notice that since $\lambda^\star \in \mathbb{R}_+^I$, the projection $x \mapsto [x]_+$ is a contraction, i.e., $\|[\lambda]_+ - \lambda^\star\| \leq \|\lambda - \lambda^\star\|$ for all $\lambda \in \mathbb{R}^I$. Using the definition of $\lambda_{k+1}, \nu_{k+1}$ from Algorithm 1, we then obtain that

$$\begin{aligned} \|\lambda_{k+1} - \lambda^\star\|^2 + \|\nu_{k+1} - \nu^\star\|^2 \leq{} & \|\lambda_k - \lambda^\star\|^2 + 2\eta_k (\lambda_k - \lambda^\star)^\top g(x_k) + \eta_k^2 \|g(x_k)\|^2 \\ & + \|\nu_k - \nu^\star\|^2 + 2\eta_k (\nu_k - \nu^\star)^\top h(x_k) + \eta_k^2 \|h(x_k)\|^2, \end{aligned} \tag{26}$$

for all $(\lambda^\star, \nu^\star) \in \Phi^\star$. To proceed, consider the conditional expectation of (26) with respect to $\mathcal{F}_k$, namely,

$$\mathbb{E}\big[\|\lambda_{k+1} - \lambda^\star\|^2 | \mathcal{F}_k\big] + \mathbb{E}\big[\|\nu_{k+1} - \nu^\star\|^2 | \mathcal{F}_k\big] \leq \|\lambda_k - \lambda^\star\|^2 + \|\nu_k - \nu^\star\|^2$$
$$+ 2\eta_k \Big[(\lambda_k - \lambda^\star)^\top \mathbb{E}_{x \sim \tilde{\mu}_k}\big[g(x)\big] + (\nu_k - \nu^\star)^\top \mathbb{E}_{x \sim \tilde{\mu}_k}\big[h(x)\big]\Big]$$
$$+ \eta_k^2 \Big[\mathbb{E}_{x \sim \tilde{\mu}_k}\|g(x)\|^2 + \mathbb{E}_{x \sim \tilde{\mu}_k}\|h(x)\|^2\Big], \quad (27)$$

where we used the fact that $\tilde{\mu}_k, \lambda_k, \nu_k \in \mathcal{F}_k$. We conclude by using the linearity of the potential energy $\mathcal{E}$ from (11) in both $\lambda$ and $\nu$ to get

$$(\lambda_k - \lambda^\star)^\top \mathbb{E}_{x \sim \tilde{\mu}_k}\big[g(x)\big] + (\nu_k - \nu^\star)^\top \mathbb{E}_{x \sim \tilde{\mu}_k}\big[h(x)\big] = \mathcal{E}(\tilde{\mu}_k, \lambda_k, \nu_k) - \mathcal{E}(\tilde{\mu}_k, \lambda^\star, \nu^\star).$$

Back in (27), we obtain

$$\mathbb{E}\big[\|\lambda_{k+1} - \lambda^\star\|^2 | \mathcal{F}_k\big] + \mathbb{E}\big[\|\nu_{k+1} - \nu^\star\|^2 | \mathcal{F}_k\big] \leq \|\lambda_k - \lambda^\star\|^2 + \|\nu_k - \nu^\star\|^2$$
$$+ 2\eta_k \big[\mathcal{E}(\tilde{\mu}_k, \lambda_k, \nu_k) - \mathcal{E}(\tilde{\mu}_k, \lambda^\star, \nu^\star)\big] \quad (28)$$
$$+ \eta_k^2 \Big[\mathbb{E}_{x \sim \tilde{\mu}_k}\|g(x)\|^2 + \mathbb{E}_{x \sim \tilde{\mu}_k}\|h(x)\|^2\Big],$$

for all $(\lambda^\star, \nu^\star) \in \Phi^\star$.

To proceed with the proof, combine (25) and (28) to get

$$W_2^2(\tilde{\mu}_{k+1/2}, \mu^\star) + \mathbb{E}\big[\|\lambda_{k+1} - \lambda^\star\|^2 | \mathcal{F}_k\big] + \mathbb{E}\big[\|\nu_{k+1} - \nu^\star\|^2 | \mathcal{F}_k\big] \leq$$
$$W_2^2(\tilde{\mu}_{k-1/2}, \mu^\star) + \|\lambda_k - \lambda^\star\|^2 + \|\nu_k - \nu^\star\|^2$$
$$+ 2\eta_k \Big[\mathcal{E}(\mu^\star, \lambda_k, \nu_k) + \mathcal{H}(\mu^\star) - \mathcal{E}(\tilde{\mu}_k, \lambda^\star, \nu^\star) - \mathcal{H}(\tilde{\mu}_k)\Big] - \eta_k m W_2^2(\tilde{\mu}_k, \mu^\star)$$
$$+ \eta_k^2 \Big[\|\nabla_{W_2} \mathcal{E}(\tilde{\mu}_k, \lambda_k, \nu_k)\|^2_{L^2(\tilde{\mu}_k)} + \mathbb{E}_{x \sim \tilde{\mu}_k}\|g(x)\|^2 + \mathbb{E}_{x \sim \tilde{\mu}_k}\|h(x)\|^2\Big]. \quad (29)$$

To upper bound the term in brackets, we add and subtract $\log(Z)$ and use the decomposition of the Lagrangian in terms of (11) to obtain

$$\mathcal{E}(\mu^\star, \lambda_k, \nu_k) + \mathcal{H}(\mu^\star) - \mathcal{E}(\tilde{\mu}_k, \lambda^\star, \nu^\star) - \mathcal{H}(\tilde{\mu}_k) = L(\mu^\star, \lambda_k, \nu_k) - L(\tilde{\mu}_k, \lambda^\star, \nu^\star).$$

Using the saddle-point property (7), we then get

$$L(\mu^\star, \lambda_k, \nu_k) - L(\tilde{\mu}_k, \lambda^\star, \nu^\star) \leq L(\mu^\star, \lambda^\star, \nu^\star) - L(\tilde{\mu}_k, \lambda^\star, \nu^\star) \leq -\,\mathrm{KL}(\tilde{\mu}_k \,\|\, \mu^\star),$$

We therefore conclude that

$$W_2^2(\tilde{\mu}_{k+1/2}, \mu^\star) + \mathbb{E}\big[\|\lambda_{k+1} - \lambda^\star\|^2 | \mathcal{F}_k\big] + \mathbb{E}\big[\|\nu_{k+1} - \nu^\star\|^2 | \mathcal{F}_k\big] \leq$$
$$W_2^2(\tilde{\mu}_{k-1/2}, \mu^\star) + \|\lambda_k - \lambda^\star\|^2 + \|\nu_k - \nu^\star\|^2 - 2\eta_k \Big[\mathrm{KL}(\tilde{\mu}_k \,\|\, \mu^\star) + \frac{m}{2} W_2^2(\tilde{\mu}_k, \mu^\star)\Big]$$
$$+ \eta_k^2 \Big[\|\nabla_{W_2} \mathcal{E}(\tilde{\mu}_k, \lambda_k, \nu_k)\|^2_{L^2(\tilde{\mu}_k)} + \mathbb{E}_{x \sim \tilde{\mu}_k}\|g(x)\|^2 + \mathbb{E}_{x \sim \tilde{\mu}_k}\|h(x)\|^2\Big]. \quad (30)$$

Since (30) holds for all $(\lambda^\star, \nu^\star) \in \Phi^\star$, it holds in particular for the minimizer of the RHS, for which we can then write $V(\tilde{\mu}_{k-1/2}, \lambda_k, \nu_k)$. By subsequently taking the minimum of the LHS, we obtain (19). ∎

*Proof of Lemma C.2.* From the updates in Algorithm 1, we obtain

$$\mathbb{E}[V(\tilde{\mu}_{1/2}, \lambda_1, \nu_1)] \leq W_2^2(\tilde{\mu}_{1/2}, \mu^\star) + \mathbb{E}[\|[\eta_0 g(x_0)]_+ - \lambda^\star\|^2] + \mathbb{E}[\|\eta_0 h(x_0) - \nu^\star\|^2]$$

for all $(\lambda^\star, \nu^\star) \in \Phi^\star$. Notice that since $\lambda^\star \in \mathbb{R}^I_+$, the projection $x \mapsto [x]_+$ is a contraction, i.e., $\|[\lambda]_+ - \lambda^\star\| \leq \|\lambda - \lambda^\star\|$ for all $\lambda \in \mathbb{R}^I$. Using the triangle inequality then yields

$$\mathbb{E}[V(\mu_{1/2}, \lambda_1, \nu_1)] \leq W_2^2(\tilde{\mu}_{1/2}, \mu_0) + W_2^2(\mu_0, \mu^\star) + \mathbb{E}[\|\eta_0 g(x_0) - \lambda^\star\|^2] + \mathbb{E}[\|\eta_0 h(x_0) - \nu^\star\|^2]$$
$$\leq W_2^2(\mu_0, \mu^\star) + \|\lambda^\star\|^2 + \|\nu^\star\|^2$$
$$+ W_2^2(\tilde{\mu}_{1/2}, \mu_0) + \eta_0^2 \mathbb{E}[\|g(x_0)\|^2] + \eta_0^2 \mathbb{E}[\|h(x_0)\|^2].$$

To proceed, observe from (18a) that $\tilde{\mu}_{1/2} = \big[\mathrm{I}_d - \eta_0 \nabla f\big]_\# \mu_0$, which implies that $W_2^2(\tilde{\mu}_{1/2}, \mu_0) \leq \eta_0^2 \|\nabla f\|^2_{L^2(\mu_0)}$. Using the bounds in Assumption 3.2, we obtain that $\mathbb{E}[V(\tilde{\mu}_{1/2}, \lambda_1, \nu_1)] \leq R_0^2$. ∎

*Proof of Proposition 3.4.* Since $c_k \geq 0$ and $\sum c_k = 1$, we can use Jensen's inequality to write

$$\left| \mathbb{E}\left[ \sum_{k=1}^{K} c_k \varphi(x_k) \right] - \mathbb{E}_{\mu^\star}[\varphi] \right| \leq \sum_{k=1}^{K} c_k \left| \int \varphi d\mu_k - \int \varphi d\mu^\star \right|.$$

Using the relation between the $\ell_1$- and $\ell_2$-norm, we further obtain

$$\sum_{k=1}^{K} c_k \left| \int \varphi d\mu_k - \int \varphi d\mu^\star \right| \leq \sqrt{\sum_{k=1}^{K} c_k \left| \int \varphi d\mu_k - \int \varphi d\mu^\star \right|^2} \tag{31}$$

If $\varphi$ is bounded by 1, then the summands on the right-hand side of (31) can be bounded by $\mathrm{TV}(\mu_k, \mu^\star)$. Indeed,

$$\left| \int \varphi d\mu_k - \int \varphi d\mu^\star \right| \leq \int \varphi |d\mu_k - d\mu^\star| \leq \int |d\mu_k - d\mu^\star| = 2\mathrm{TV}(\mu_k, \mu^\star).$$

The total variation distance can in turn be bounded by the KL divergence using Pinsker's inequality. We therefore obtain

$$\left| \mathbb{E}\left[ \sum_{k=1}^{K} c_k \varphi(x_k) \right] - \mathbb{E}_{\mu^\star}[\varphi] \right| \leq \sqrt{2 \sum_{k=1}^{K} c_k \, \mathrm{KL}(\mu_k \| \mu^\star)} \leq \sqrt{2\Delta_K}.$$

On the other hand, if $\varphi$ is 1-Lipschitz, the summands on the right-hand side of (31) are bounded by

$$\left| \int \varphi d\mu_k - \int \varphi d\mu^\star \right|^2 \leq W_1^2(\mu_k, \mu^\star) \leq W_2^2(\mu_k, \mu^\star),$$

which implies that

$$\left| \mathbb{E}\left[ \sum_{k=1}^{K} c_k \varphi(x_k) \right] - \mathbb{E}_{\mu^\star}[\varphi] \right| \leq \sqrt{\frac{2\Delta_K}{m}},$$

as long as $m > 0$. $\blacksquare$

# D  Proofs from Section 3.2

*Proof of Theorem 3.6.* The proof is based on the analysis of the stochastic dual ascent algorithm

$$\lambda_{k+1} = \left[\lambda_k + \eta_k g(\xi_k)\right]_+, \quad \text{for } \xi_k \sim \bar{\mu}_k \text{ such that } \text{KL}(\bar{\mu}_k \| \mu_{\lambda_k}) \leq \epsilon, \tag{32}$$

where $\mu_{\lambda_k}$ is the Lagrangian minimizer from (4) and $\epsilon > 0$. Observe that, once again, we analyze the conditional distribution $\bar{\mu}_k = \mathcal{L}(\xi_k | \lambda_k)$. We collect this result in the following proposition:

**Proposition D.1.** *Consider the iterations (32) and assume that $\mathbb{E}_\mu[g_i^2] \leq G^2$ for all $\mu \in \mathcal{P}_2(\mathbb{R}^d)$. Then, for $0 < \eta_k \leq \eta$ and $\epsilon \leq \eta G^2$, there exists $B < \infty$ such that*

$$\frac{1}{K} \sum_{k=0}^{K-1} \mathbb{E}[\text{KL}(\bar{\mu}_k \| \mu^\star)] \leq \epsilon + \frac{\eta G^2}{2} + \frac{2B^2 I}{\eta K}. \tag{33}$$

*The expectations are taken over the samples $\xi_k \sim \bar{\mu}_k$.*

We conclude by combining Prop. D.1 with [29, Theorem 1], which characterizes the convergence of the LMC algorithm (3) under Assumption 3.5. Indeed, using the $\gamma_k, N_k^0$ from Theorem 3.6 in Algorithm 1 guarantees that the law $\bar{\mu}_k$ of $x_k | \lambda_k$ is such that $\text{KL}(\bar{\mu}_k \| \mu_{\lambda_k}) \leq \epsilon$, i.e., satisfies the conditions in Prop. D.1. We can then apply Jensen's inequality to get that $\mathbb{E}[\text{KL}(\bar{\mu}_k \| \mu^\star)] \geq \text{KL}(\mathbb{E}[\bar{\mu}_k] \| \mu^\star) = \text{KL}(\mu_k \| \mu^\star)$, where $\mu_k$ is the law of $x_{N_k^0}$ in Algorithm 2. ∎

*Proof of Prop. D.1.* The proof relies on the following lemmata:

**Lemma D.2.** *For all $\mu \in \mathcal{P}_2(\mathbb{R}^d)$ such that $\text{KL}(\mu \| \mu_\lambda) \leq \epsilon$, the expected value $E_\mu[g]$ is an approximate subgradients of the dual function $d$ in (6) at $\lambda \in \mathbb{R}_I^+$, i.e.,*

$$d(\lambda) \geq d(\lambda') + (\lambda - \lambda')^\top E_\mu[g] - \epsilon, \quad \text{for all } \lambda' \in \mathbb{R}_I^+. \tag{34}$$

**Lemma D.3.** *Under the conditions of Prop. D.1, it holds for all $k$ that*

$$\|\lambda^\star\|_1 \leq B_0 \triangleq \frac{C - D^\star + \eta G^2 + \epsilon}{\delta} \quad \text{and} \quad \mathbb{E}[\|\lambda_k - \lambda^\star\|^2] \leq B^2 \triangleq 2B_0^2 + 3\eta^2 G^2.$$

**Lemma D.4.** *The sequence $(\lambda_k, g(\xi_k))$ obtained from (32) is such that*

$$\frac{1}{K} \sum_{k=0}^{K=1} \mathbb{E}\left[\lambda_k^\top g(\xi_k)\right] \geq -\frac{\eta G^2}{2}, \tag{35}$$

*where the expectation is taken over realizations of $\xi_k$.*

Before proving these results, let us show how they imply Prop. D.1. Start by noticing from (32) that $\lambda_{i,k+1} \geq \lambda_{i,k} + \eta g_i(\xi_k)$. Solving the recursion and recalling that $\lambda_0 = 0$ then yields

$$\lambda_{i,K} \geq \eta \sum_{k=0}^{K-1} g_i(\xi_k).$$

Taking the expected value over $\xi_k$ and dividing by $\eta K$, we obtain

$$\frac{1}{K} \sum_{k=0}^{K-1} \mathbb{E}_{\bar{\mu}_k}[g_i] \leq \frac{\mathbb{E}[\lambda_{i,K}]}{\eta K} \leq \frac{\mathbb{E}[|\lambda_{i,K} - \lambda_i^\star|] + \lambda_i^\star}{\eta K}, \tag{36}$$

where the last bound stems from the triangle inequality. Since the upper bound is non-negative for all $i$, we use the fact that the maximum of a set of values is less than the sum of those values to write

$$\max_{i=1,\dots,I} \left[\frac{1}{K} \sum_{k=0}^{K-1} \mathbb{E}_{\bar{\mu}_k}[g_i]\right] \leq \frac{\mathbb{E}[\|\lambda_K - \lambda^\star\|_1] + \|\lambda^\star\|_1}{\eta K} \leq \frac{(1 + \sqrt{I})B}{\eta K}, \tag{37}$$

where we used Lemma D.3 together with $(\mathbb{E}[\|z\|_1])^2 \leq I \cdot \mathbb{E}[\|z\|^2]$ and $B_0 < B$. Observe that (37) bounds the infeasibility of the ergodic average $\frac{1}{K} \sum_{k=0}^{K-1} \bar{\mu}_k$ for $\bar{\mu}_k$ as in (32).

To proceed, use the relation between the normalization of $\mu_\lambda$ and the dual function value [see (6)] to decompose the KL divergence between $\bar{\mu}_k$ and $\mu^\star$ as

$$\mathrm{KL}(\bar{\mu}_k\|\mu^\star) = \mathrm{KL}(\bar{\mu}_k\|\mu_{\lambda_k}) + (\lambda^\star - \lambda_k)^\top \mathbb{E}_{\bar{\mu}_k}[g] + d(\lambda_k) - d(\lambda^\star).$$

Since $\mathrm{KL}(\bar{\mu}_k\|\mu_{\lambda_k}) \leq \epsilon$ and $d(\lambda) \leq d(\lambda^\star)$ for all $\lambda \in \mathbb{R}_+^I$, we get

$$\mathrm{KL}(\bar{\mu}_k\|\mu^\star) \leq \epsilon + (\lambda^\star - \lambda_k)^\top \mathbb{E}_{\bar{\mu}_k}[g]$$

Averaging over $k$ and using Hölder's inequality then yields

$$\frac{1}{K} \sum_{k=0}^{K-1} \mathrm{KL}(\bar{\mu}_k\|\mu^\star) \leq \epsilon + \|\lambda^\star\|_1 \cdot \max_{i=1,\ldots,I} \left[ \frac{1}{K} \sum_{k=0}^{K-1} \mathbb{E}_{\bar{\mu}_k}[g_i] \right] - \frac{1}{K} \sum_{k=0}^{K-1} \lambda_k^\top \mathbb{E}_{\bar{\mu}_k}[g],$$

which from Lemma D.3 and (37) becomes

$$\frac{1}{K} \sum_{k=0}^{K-1} \mathrm{KL}(\bar{\mu}_k\|\mu^\star) \leq \epsilon + \frac{\left(1 + \sqrt{I}\right)B^2}{\eta K} - \frac{1}{K} \sum_{k=0}^{K-1} \lambda_k^\top \mathbb{E}_{\bar{\mu}_k}[g],$$

where we used $B_0 < B$. Taking the expected value, applying Lemma D.4, and taking $1 + \sqrt{I} \leq 2I$ for $I \geq 1$ yields (33). ∎

### D.1 Proof of Lemmata D.2–D.4

*Proof of Lemma D.2.* From the definition of the Lagrangian (4) and the dual function (6) (with $J = 0$), we obtain $L(\mu, \lambda) = \mathrm{KL}(\mu\|\mu_\lambda) + d(\lambda)$. Using the fact that $\mathrm{KL}(\mu\|\mu_\lambda) \leq \epsilon$, we get

$$0 \leq d(\lambda) - L(\mu, \lambda) + \epsilon. \tag{38}$$

Again using the definition of the dual function (6), we also obtain that $d(\lambda') \leq L(\mu, \lambda')$. Adding to (38) then gives

$$d(\lambda') \leq +d(\lambda) + L(\mu, \lambda') - L(\mu, \lambda) + \epsilon. \tag{39}$$

Notice from (4) that the first term of the Lagrangians in (39) cancel out, leading to (34). ∎

*Proof of Lemma D.3.* Start by combining the update in (32) and the fact that, since $\lambda^\star \in \mathbb{R}_+^I$, the projection $x \mapsto [x]_+$ is a contraction, to get

$$\|\lambda_{k+1} - \lambda^\star\|^2 \leq \|\lambda_k - \lambda^\star\|^2 + 2\eta(\lambda_k - \lambda^\star)^\top g(\xi_k) + \eta_k^2 \|g(\xi_k)\|^2.$$

Taking the conditional expectation given $\lambda_k$ then yields

$$\mathbb{E}[\|\lambda_{k+1} - \lambda^\star\|^2 \mid \lambda_k] \leq \|\lambda_k - \lambda^\star\|^2 + 2\eta(\lambda_k - \lambda^\star)^\top \mathbb{E}[g(\xi_k) \mid \lambda_k] + \eta^2 G^2.$$

where we used the fact that $\mathbb{E}_\mu[\|g\|^2] \leq G^2$ for any $\mu$. Noticing from (32) that $\mathbb{E}[g(\xi_k) \mid \lambda_k] = \mathbb{E}_{\bar{\mu}_k}[g]$, we can use Lemma D.2 to get

$$\mathbb{E}[\|\lambda_{k+1} - \lambda^\star\|^2 \mid \lambda_k] \leq \|\lambda_k - \lambda^\star\|^2 + 2\eta\Big[d(\lambda_k) - D^\star + \eta G^2 + \epsilon\Big], \tag{40}$$

where we used the fact that $\eta/2 < \eta$ to simplify the relation.

To proceed, consider the set of approximate maximizers of the dual function

$$\mathcal{D} \triangleq \left\{ \lambda \in \mathbb{R}_+^I \mid d(\lambda) \geq D^\star - \eta G^2 - \epsilon \right\}. \tag{41}$$

Notice that $\Phi^\star \subseteq \mathcal{D}$. Since there exists at least one $\lambda^\star$ that achieves $D^\star$ (Prop. 2.2), $\mathcal{D}$ is not empty. Notice that for $\lambda_k \notin \mathcal{D}$, we have that $d(\lambda_k) - D^\star + \eta G^2 + \epsilon < 0$. By the towering property, we therefore obtain from (40) that

$$\mathbb{E}[\|\lambda_{k+1} - \lambda^\star\|^2 \mid \lambda_k \notin \mathcal{D}] < \mathbb{E}[\|\lambda_k - \lambda^\star\|^2 \mid \lambda_k \notin \mathcal{D}], \tag{42}$$

since $\eta > 0$.

To bound the case when $\lambda_k \in \mathcal{D}$ we use the strictly feasible candidate $\mu^\dagger$ from Assumption 2.1. Indeed, recall that $\mathrm{KL}(\mu^\dagger\|\pi) \leq C$ and $E_{\mu^\dagger}[g_i] \leq -\delta < 0$ for all $i$. From the definition of the dual function (6), we obtain that

$$d(\lambda) \leq L(\mu^\dagger, \lambda) \leq C - \|\lambda\|_1 \delta, \tag{43}$$

where we used the fact that $\lambda \in \mathbb{R}_+^I$ to write $\sum_i \lambda_i = \|\lambda\|_1$. Hence, it follows that

$$\|\lambda\|_1 \leq B_0 \triangleq \frac{C - D^\star + \eta G^2 + \epsilon}{\delta}, \quad \text{for all } \lambda \in \mathcal{D}. \tag{44}$$

Using the fact that $\|z\|^2 \leq \|z\|_1^2$ for all $z$ and that $\Phi^\star \subset \mathcal{D}$, we immediately obtain that $\|\lambda - \lambda^\star\|^2 \leq 2B_0^2$ for all $\lambda \in \mathcal{D}$. Using the towering property and the fact that $D^\star \geq d(\lambda_k)$ and $\epsilon \leq \eta G^2$ yields

$$\mathbb{E}[\|\lambda_{k+1} - \lambda^\star\|^2 \mid \lambda_k \in \mathcal{D}] \leq 2B_0^2 + 2\eta\epsilon + \eta^2 G^2 \leq B^2. \tag{45}$$

To conclude, we write

$$\mathbb{E}[\|\lambda_{k+1} - \lambda^\star\|^2] = \mathbb{E}[\|\lambda_{k+1} - \lambda^\star\|^2 \mid \lambda_k \in \mathcal{D}] \, \mathbb{P}[\lambda_k \in \mathcal{D}]$$
$$+ \mathbb{E}[\|\lambda_{k+1} - \lambda^\star\|^2 \mid \lambda_k \notin \mathcal{D}] \, \mathbb{P}[\lambda_k \notin \mathcal{D}].$$

Using (42) and (45) then yields

$$\mathbb{E}[\|\lambda_{k+1} - \lambda^\star\|^2] \leq B_1^2 \, \mathbb{P}[\lambda_k \in \mathcal{D}] + \mathbb{E}[\|\lambda_k - \lambda^\star\|^2 \mid \lambda_k \notin \mathcal{D}] \, \mathbb{P}[\lambda_k \notin \mathcal{D}]$$
$$\leq \max(B^2, \mathbb{E}[\|\lambda_k - \lambda^\star\|^2 \mid \lambda_k \notin \mathcal{D}]).$$

Since both (42) and (45) holds independently of $\lambda_{k+1}$, we can also write

$$\mathbb{E}[\|\lambda_k - \lambda^\star\|^2 \mid \lambda_k \notin \mathcal{D}] \leq \max(B^2, \mathbb{E}[\|\lambda_{k-1} - \lambda^\star\|^2 \mid \lambda_{k-1} \notin \mathcal{D}])$$

Applying these relations recursively, we obtain that

$$\mathbb{E}[\|\lambda_{k+1} - \lambda^\star\|^2] \leq \max(B^2, \|\lambda_0 - \lambda^\star\|^2) = \max(B_1^2, \|\lambda^\star\|^2).$$

Noticing from (44) that since $\lambda^\star \in \mathcal{D}$ we have $\|\lambda^\star\|^2 \leq B_0^2 < B^2$ then concludes the proof. ∎

*Proof of Lemma D.4.* To bound (35), we once again use the non-expansiveness of the projection to obtain

$$\|\lambda_{k+1}\|^2 \leq \|\lambda_k\|^2 + \eta^2 \|g(\xi_k)\|^2 + 2\eta \lambda_k^\top g(\xi_k).$$

Taking the expectation and using the fact that $\mathbb{E}_\mu[\|g\|^2] \leq G^2$, we get

$$\mathbb{E}[\|\lambda_{k+1}\|^2] \leq \mathbb{E}[\|\lambda_k\|^2] + \eta^2 G^2 + 2\eta \, \mathbb{E}[\lambda_k^\top g(\xi_k)].$$

Applying this relation recursively from $K$ and using the fact that $\lambda_0 = 0$ (deterministic) yields

$$\mathbb{E}[\|\lambda_K\|^2] \leq K\eta^2 G^2 + 2\eta \sum_{k=0}^{K-1} \mathbb{E}[\lambda_k^\top g(\xi_k)].$$

Since $\mathbb{E}[\|\lambda_K\|^2] \geq 0$, we can divide by $2\eta K$ to obtain the desired result. ∎

# E  Applications

In this section, we provide further details on the example applications described in Section 2.2 as well as additional results from the experiments in Section 4. In these experiments, we start all chains at zero (unless stated otherwise) and use different step-sizes for each of the updates in steps 3–5 from Algorithm 1. We refer to them as $\eta_x$, $\eta_\lambda$, and $\eta_\nu$. In contrast, we do not use diminishing step-sizes.

## E.1  Sampling from convex sets

We are interested in sampling from target distribution

$$\pi^o(x) \propto e^{-f(x)} \mathbb{I}(x \in \mathcal{C}), \tag{46}$$

for some closed, convex set $\mathcal{C} \subset \mathbb{R}^d$. Several methods have been developed to tackle this problem, based on projections [13, 64], mirror maps [12, 15, 16], and barriers [17, 18]. Here, we consider a constrained sampling approach based on (PI) instead.

To do so, note that sampling from (46) is equivalent to sampling from

$$\mu^\star \in \operatorname*{argmin}_{\mu \in \mathcal{P}_2(\mathcal{C})} \quad \mathrm{KL}(\mu \| \pi)$$

for the unconstrained $\pi \propto e^{-f} \in \mathcal{P}_2(\mathbb{R}^d)$. Note that this is exactly (PII). Now let $\mathcal{C}$ be described by the intersection of the 0-sublevel sets of convex functions $\{s_i\}_{i=1,\dots,I}$, i.e.,

$$\mathcal{C} = \bigcap_{i=1}^{I} \{x : s_i(x) \le 0\}.$$

Such a description is always possible by, e.g., considering the distance function $d(x, \mathcal{C}) = \inf_{y \in \mathcal{C}} \|x - y\|_2$, which is convex (since $\mathcal{C}$ is convex) and for which $\{x : d(x, \mathcal{C}) \le 0\} = \mathcal{C}$ (since $\mathcal{C}$ is closed). Immediately, we see that solving (PII) is equivalent to solving

$$\begin{aligned} \min_{\mu \in \mathcal{P}_2(\mathbb{R}^d)} \quad & \mathrm{KL}(\mu \| \pi) \\ \text{subject to} \quad & \mathbb{E}_{x \sim \mu}\big[[s_i(x)]_+\big] \le 0, \ i = 1, \dots, I, \end{aligned} \tag{PVI}$$

where $[z]_+ = \max(0, z)$. Note that (PVI) has the same form as (PI).

To see why this is the case, consider without loss of generality that $i = 1$. Since $[s_i(x)]_+ \ge 0$ for all $x$ by definition, it immediately holds that

$$\mathcal{C} = \{x : s(x) \le 0\} = \{x : [s(x)]_+ \le 0\} = \{x : [s(x)]_+ = 0\}.$$

By the monotonicity of Lebesgue integration, we obtain that the feasibility set of (PVI) is

$$\begin{aligned} \mathcal{F} &= \Big\{\mu \in \mathcal{P}_2(\mathbb{R}^d) : \mathbb{E}_{x \sim \mu}\big[[s(x)]_+\big] \le 0\Big\} \\ &= \Big\{\mu \in \mathcal{P}_2(\mathbb{R}^d) : \mathbb{E}_{x \sim \mu}\big[[s(x)]_+\big] = 0\Big\} \\ &= \Big\{\mu \in \mathcal{P}_2(\mathbb{R}^d) : [s(x)]_+ = 0, \ \mu\text{-a.e.}\Big\}. \end{aligned} \tag{47}$$

In other words, the feasibility set of (PVI) is in fact $\mathcal{F} = \{\mu \in \mathcal{P}_2(\mathbb{R}^d) : \mu(\mathcal{C}) = 1\} = \mathcal{P}_2(\mathcal{C})$.

To illustrate the use of (PVI), consider the one-dimensional truncated Gaussian sampling problem from Section 4. Namely, we wish to sample from a standard Gaussian distribution $\mathcal{N}(0, 1)$ truncated to $\mathcal{C} = [1, 3]$. In the language of (PVI), we take $f(x) = x^2/2$ (i.e., $\pi \propto e^{-x^2/2}$) and $s_i(x) =$

Table 2: Mean and variance estimates

|  | True mean | Proj. LMC | Mirror LMC | PD-LMC |
|---|---|---|---|---|
| 1D truncated Gaussian | 1.510 | 1.508 | 1.470 | 1.488 |
| 2D truncated Gaussian | $[0.368, 0.368]$ | $[0.611, 0.610]$ | $[0.312, 0.418]$ | $[0.446, 0.444]$ |

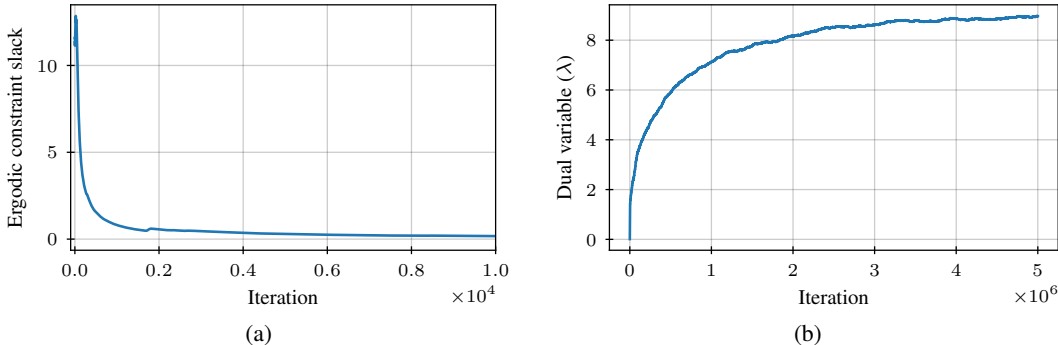

Figure 5: One-dimensional truncated Gaussian sampling: (a) Ergodic average of the constraint function (slack) and (b) Evolution of the dual variable $\lambda$.

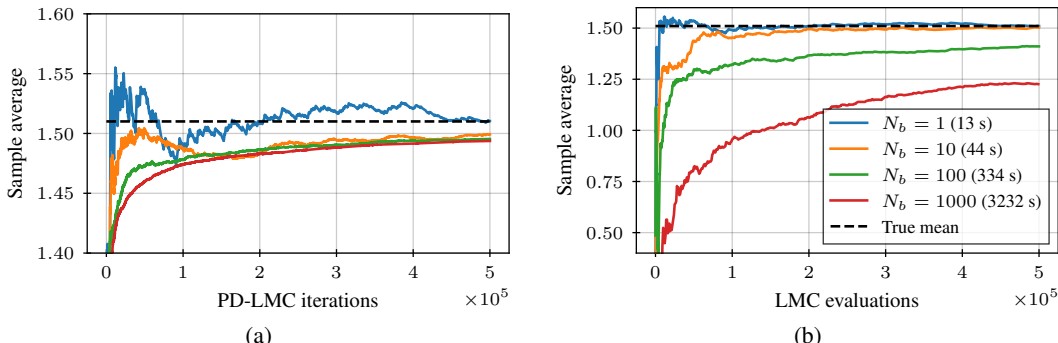

Figure 6: The effect of the mini-batch size $N_b$ on PD-LMC for sampling from a 1D truncated Gaussian: Estimated mean vs. (a) iteration and (b) LMC evaluations.

$(x-1)(x-3)$. In order to satisfy the assumptions of our convergence guarantees (particularly 2.1), we leave some slack in the constraints by considering $\mathbb{E}_\mu[[s_i(x)]_+] \leq 0.005$. This also helps with the numerical stability of the algorithm. Fig. 1 shows histograms of the samples obtained using PD-LMC (Algorithm 1 with $\eta_x = \eta_\lambda = 10^{-3}$), the projected LMC (Proj. LMC, $\eta = 10^{-3}$) from [13], and the mirror LMC ($\eta = 10^{-3}$) from [59]. In all cases, we take $5 \times 10^6$ samples and keep only the second half.

Observe that, due to the projection step, Proj. LMC generates an excess of samples close to the boundaries. In fact, it generates over three times more than required. This leads to an underestimation of the distribution mean and variance (Table 2). A similar effect is observed for mirror LMC. In contrast, PD-LMC provides a more accurate estimate. Nevertheless, PD-LMC imposes constraints on the distribution $\mu$ rather than its samples. Indeed, note from (47) that its feasibility set is such that samples belong to $\mathcal{C}$ *almost surely*, which still allows for a (potentially infinite) number of realizations outside of $\mathcal{C}$. Yet, though PD-LMC is not an *interior-point method*, Theorems 3.3–3.6 show that excursions of iterates outside of $\mathcal{C}$ become less frequent as the algorithm progresses. We can confirm this is indeed the case in Fig. 5a, which shows the ergodic average of $[s(x)]_+$ along the samples of PD-LMC. Note that it almost vanishes by iteration $10^4$ even though the dual variable $\lambda$ only begins to stabilize later (Fig. 5b). This is not surprising given that it is guaranteed by Prop. 3.4. In fact, only roughly $2\%$ of the samples displayed in Fig. 1 are not in $\mathcal{C}$.

Before proceeding, we examine whether the convergence of PD-LMC could be improved by averaging more than one LMC samples when updating the dual variables, i.e., using mini-batches in steps 4–5 of Algorithm 1. Mini-batches will reduce the variance of the dual updates, although at the cost of additional LMC steps per iteration. To compensate for this fact, Fig. 6b displays the evolution of the ergodic average of PD-LMC samples as a function of the number of LMC evaluations rather than the number of iterations (as in Fig. 6a). Notice that, in this application, increasing the number of LMC samples $N_b$ does not lead to faster convergence. This illustrates that, though mini-batches could be useful in some applications (particularly when the constraints are *not* convex, as in Section 3.2), it

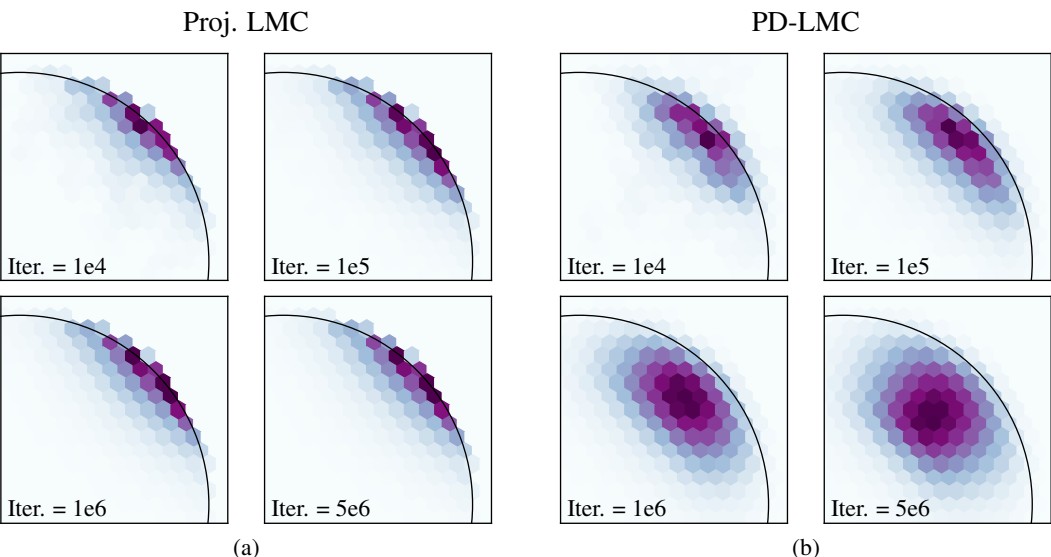

Figure 7: Density estimate of two-dimensional truncated Gaussian using samples from (a) Proj. LMC and (b) PD-LMC.

is not immediate that their benefits always outweigh the increased computational cost. Oftentimes, using a single LMC sample is more than enough. It is worth noting that using PD-LMC with a large mini-batch $N_b$ was suggested in [24] to approximate the expectation needed by their continuous-time algorithm. As we see here, this is neither necessary nor always beneficial.

We now turn to a more challenging, two-dimensional applications. We seek to sample from a Gaussian located at $[2, 2]$ with covariance $\mathrm{diag}([1, 1])$ restricted to an $\ell_2$-norm unit ball (Fig. 1). Specifically, we use $f(x) = \|x\|^2 / 2$ (i.e., $\pi \propto e^{-\|x\|^2/2}$) and $s_i(x) = \|x\|^2 - 1$. Once again, we leave some slack to the algorithm by taking the constraint in (PVI) to be $\mathbb{E}_\mu[[s_i(x)]_+] \leq 0.001$. For reference, we also display samples from the real distribution obtained using rejection sampling.

This is indeed a challenging problem. The boundary of $\mathcal{C}$ is 2 standard deviations away from the mean of the target distribution, which means that samples from the target $\pi$ are extremely scarce this region. Indeed, using the untruncated Gaussian as a proposal for rejection sampling yields an acceptance rate of approximately $1\%$. The strong push of the potential $f$ towards the exterior of $\mathcal{C}$ leads Proj. LMC ($\eta = 10^{-3}$; the last $10^6$ samples are used after running $5 \times 10^6$ iterations) to be now even more concentrated around its boundary. In fact, almost $25\%$ of its samples are in an annular region of radius $[0.999, 1)$, where only $0.14\%$ of the samples should be according to rejection sampling. Indeed, note from Fig. 7a, that even as iterations advance, the samples of Proj. LMC continue concentrate close to the boundary.

In contrast, PD-LMC ($\eta_x = 10^{-3}$ and $\eta_\lambda = 2 \times 10^{-1}$) only place $1.8\%$ of its samples outside of $\mathcal{C}$, mostly during the initial phase of the algorithm (Fig. 7a). Indeed, the average of the constraint function along samples from PD-LMC essentially vanishes around iteration $5 \times 10^4$. Achieving this requires larger values of $\lambda$ (on the order of 250, Fig. 8b) compared to the one-dimensional case (Fig. 5b). This reflects the difficulty of constraining samples to $\mathcal{C}$ in this instance, a statement formalized in the perturbation results of Prop. 2.2(iv). Due to the more amenable numerical properties of the barrier function, mirror LMC ($\eta = 10^{-3}$) performs well without concentrating samples on the boundary ($0.15\%$ of the samples on the annular region of radius $[0.999, 1)$).

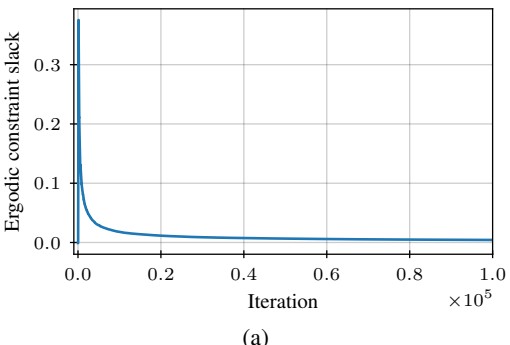 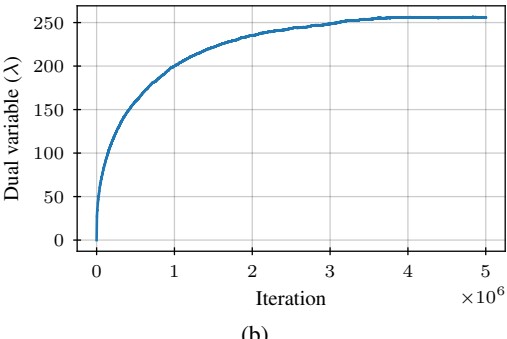

(a)            (b)

Figure 8: Two-dimensional truncated Gaussian sampling: (a) Ergodic average of the constraint function (slack) and (b) Evolution of the dual variable $\lambda$.

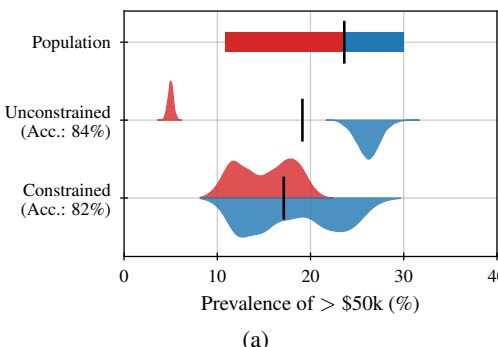 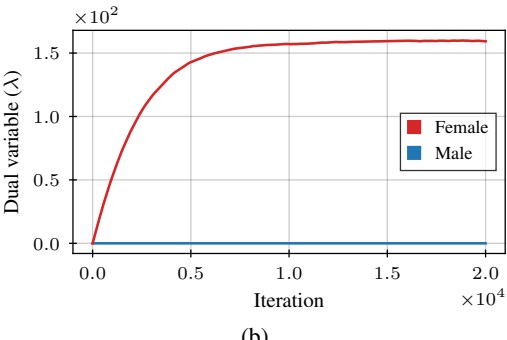

(a)            (b)

Figure 9: Fair Bayesian logistic regression on the Adult dataset: (a) prevalence of positive outputs and (b) dual variables.

## E.2    Rate-constrained Bayesian models

While rate constraints have become popular in ML due to their central role in fairness (see, e.g., [20]), they find applications in robust control [76–78] and to express other requirements on the confusion matrix, such as precision, recall, and false negatives [21]. For illustration, we consider here the problem of fairness in Bayesian classification.

Let $q(x;\theta) = \mathbb{P}[y = 1|x,\theta]$ denote the probability of a positive outcome ($y = 1$) given the observed features $x \in \mathcal{X}$ and the parameters $\theta$ distributed according to the posterior $\pi$. This posterior is determined, e.g., by some arbitrary Bayesian model based on observations $\{(x_n, y_n)\}_{n=1,\dots,N}$. Hence, $\mathbb{E}_{\theta \sim \pi}[q(x;\theta)]$ denotes the likelihood of a positive outcome for $x$. Consider now a protected group, represented by a measurable subset $\mathcal{G} \subset \mathcal{X}$, for which we wish to enforce statistical parity. In other words, we would like the prevalence of positive outcomes to be roughly the same as that of the whole population. Thus, we now want to sample not from the posterior $\pi$, but from a close-by distribution of parameters $\theta$ that ensures this parity. Explicitly, for some tolerance $\delta > 0$, we want to sample from

$$\mu^\star \in \min_{\mu \in \mathcal{P}_2(\mathbb{R}^d)} \quad \mathrm{KL}(\mu \| \pi)$$
$$\text{subject to} \quad \mathbb{E}_{x,\theta \sim \mu}\big[q(x;\theta) \mid \mathcal{G}\big] \geq \mathbb{E}_{x,\theta \sim \mu}\big[q(x;\theta)\big] - \delta. \tag{PVII}$$

Naturally, we can account for more than one protected by incorporating additional constraints.

In our experiments, we take $\pi$ to be the posterior of a Bayesian logistic regression model for the Adult dataset from [60] (details on data pre-processing can be found in [61]). The $N = 32561$ data points in the training set are composed of $d = 62$ socio-economical features ($x \in \mathbb{R}^d$, including the intercept) and the goal is to predict whether the individual makes more than US\$ 50000 per year ($y \in \{0, 1\}$). The posterior is obtained by combining a binomial log-likelihood with independent zero-mean Gaussian (log)priors ($\sigma^2 = 3$) on each parameter of the model, i.e., we consider the

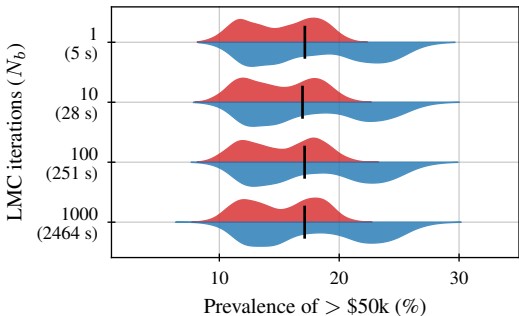

Figure 10: The effect of the mini-batch size $N_b$ on PD-LMC in fair Bayesian classification.

potential

$$f(\beta) = \sum_{n=1}^{N} \log(1 + e^{-(2y_n - 1)x_n^\top \theta}) + \sum_{i=0}^{d} \frac{\theta_i^2}{2\sigma^2}. \tag{48}$$

We begin by using the LMC algorithm from (3) (i.e., we impose no constraints) to collect samples of the coefficients $\theta$ from this posterior ($\eta = 10^{-4}$; the last $10^4$ samples are used after running $2 \times 10^4$ iterations). We find that, while the probability of positive outputs is $19.1\%$ across the whole test set, it is $26.2\%$ among males and $0.05\%$ among females. Looking at the distribution of this probability over the unconstrained posterior $\pi$ (Fig. 9a), we see that this behavior goes beyond the mean. The model effectively amplifies the inequality already present in the test set, where the prevalence of positive outputs is $30.6\%$ among males and $10.9\%$ among females.

To overcome this disparity, we consider *gender* to be the protected class in (PVII), constraining both $\mathcal{G}_{\text{female}}$ and $\mathcal{G}_{\text{male}}$. We formulate the constraint of (PVII) using an empirical distribution induced from the data. Explicitly, we consider constraints

$$\frac{1}{|\mathcal{G}_{\text{female}}|} \sum_{n \in \mathcal{G}_{\text{female}}} \mathbb{E}_{\theta \sim \mu}\big[q(x_n; \theta)\big] \geq \frac{1}{N} \sum_{n=1}^{N} \mathbb{E}_{\theta \sim \mu}\big[q(x_n; \theta)\big] - \delta$$

$$\frac{1}{|\mathcal{G}_{\text{male}}|} \sum_{n \in \mathcal{G}_{\text{male}}} \mathbb{E}_{\theta \sim \mu}\big[q(x_n; \theta)\big] \geq \frac{1}{N} \sum_{n=1}^{N} \mathbb{E}_{\theta \sim \mu}\big[q(x_n; \theta)\big] - \delta$$

where $\mathcal{G}_{\text{female}}, \mathcal{G}_{\text{male}} \subseteq \{1, \ldots, N\}$ partition the data set. For these experiments, we take $\delta = 0.01$. Using PD-LMC ($\eta_x = 10^{-4}$, $\eta_\lambda = 5 \times 10^{-3}$), we then obtain a new set of samples from the logistic regression parameters $\theta$ that lead to a prevalence of positive outcomes (in the test set) of $17.1\%$ over the whole population, $18.1\%$ for males, and $15.1\%$ for females. In fact, we notice a substantial overlap between the distributions of this probability across the constrained posterior $\mu^\star$ for male and female (Fig. 9a). Additionally, this substantial improvement over the previously observed disparity comes at only a minor reduction in accuracy. Though both distributions change considerably, notice from the value of the dual variables that these changes are completely guided by the *female* group. Indeed, $\lambda_{\text{male}} = 0$ throughout the execution of PD-LMC (Fig. 9b).

Before proceeding, we once again examine the effect of using multiple LMC samples to update the dual variables, i.e., using mini-batches in steps 4–5 of Algorithm 1. Fig. 10 shows the distribution of the prevalence of positive predictions ($> \$50\text{k}$) for different mini-batch sizes $N_b$. In all cases, we collect $2 \times 10^4$ samples, which means that we evaluate $2N_b \times 10^4$ LMC updates (step 3 in Algorithm 1). Same as in the 1D truncated Gaussian case, we notice no difference between the resulting distributions. This is to be expected given our results (Theorem 3.3). The computation time, on the other hand, increases considerably with the mini-batch size. Once again, we note that PD-LMC with a large mini-batch $N_b$ was used in the experiments of [24] to overcome the challenge of computing an expectation in their dual variable updates. In turns out that this computationally intensive modification is not necessary.

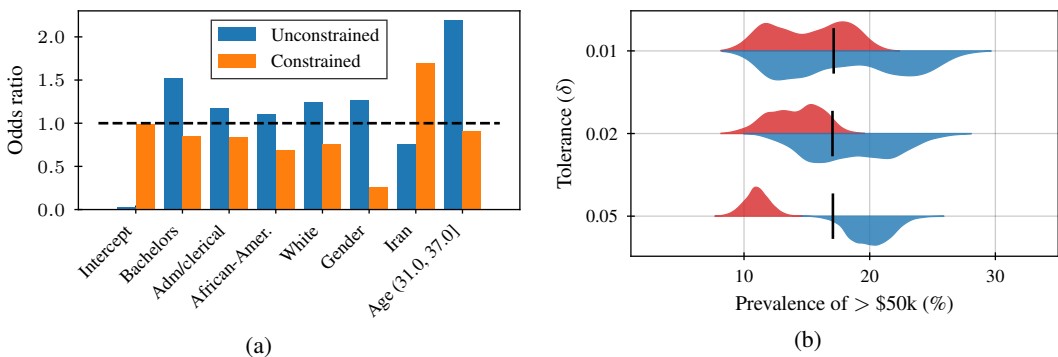

Figure 11: Counterfactual sampling in fair Bayesian logistic regression: (a) Selected (mean) coefficients and (b) different tolerances.

### E.3 Counterfactual sampling

Previous applications were primarily interested in *sampling from* $\mu^\star$, the constrained version of the target distribution $\pi$. The goal of *counterfactual sampling*, on the other hand, is to *probe* the probabilistic model $\pi$ by evaluating its compatibility with a set of moment conditions. It is therefore interested not only in $\mu^\star$, but in how each condition affects the value $P^\star = \text{KL}(\mu^\star \| \pi)$. We next describe how constrained sampling can be used to tackle this problem.

Let $\pi$ denote a reference probabilistic model, such as the posterior of the Bayesian logistic model in (48). Standard Bayesian hypothesis tests can be used to evaluate the validity of *actual statements* such as "is it true that $\mathbb{E}_{x\sim\pi}[g(x)] \leq 0$? or $\mathbb{E}_{x\sim\pi}[h(x)] = 0$?" Hence, we could check "is $\pi$ more likely to yield a positive output for a male than a female individual?" (from the distributions under *Unconstrained* in Fig. 9a, this is probably the case). In contrast, counterfactual sampling is concerned with *counterfactual statements* such as "how would the world have been if $\mathbb{E}[g(x)] \leq 0$?" In the case of fairness, "how would the model have been if it predicted positive outcomes more equitably?"

Constrained sampling evaluates these counterfactual statements in two ways. First, by providing realizations of this alternative, counterfactual world ($\mu^\star$). For instance, we can inspect the difference between realizations of $\pi$, obtained using the traditional LMC in (3), and $\mu^\star$, obtained using PD-LMC (Algorithm 1). In Fig. 11a, we show the mean of some coefficients of the Bayesian logistic models from Section E.2. Notice that it is not enough to normalize the Intercept and reduce the advantage given to males (*Female* is encoded as Male $= 0$). This alternative model also compensates for other correlated features, such as education (*Bachelor*), profession (*Adm/clerical*), and age.

Second, constrained sampling evaluates the "compatibility" of each counterfactual condition (constraint) world with the reference model. While Algorithm 1 does not evaluate $P^\star$ explicitly, it provides measures of its sensitivity to perturbations of the constraints: the Lagrange multipliers $(\lambda^\star, \nu^\star)$. Indeed, recall from Prop. 2.2 that

$$\mu^\star \propto \pi \times \left( \prod_{i=1}^{I} e^{-\lambda_i^\star g_i} \right) \times \left( \prod_{j=1}^{J} e^{-\nu_j^\star h_j} \right).$$

Hence, $(\lambda^\star, \nu^\star)$ describe the magnitude of *tilts* needed for the reference model $\pi$ to satisfy the conditions $\mathbb{E}[g(x)] \leq 0$ or $\mathbb{E}[h(x)] = 0$. This relation is made explicit in Prop. 2.2(iv).

Concretely, observe that the dual variable relative to the constraint on the male subgroup is always zero (Fig. 9b). This implies that $\pi$ is fully compatible with the condition

$$\frac{1}{|\mathcal{G}_{\text{male}}|} \sum_{n \in \mathcal{G}_{\text{male}}} \mathbb{E}_{\theta \sim \mu} \big[ q(x_n; \theta) \big] \geq \frac{1}{N} \sum_{n=1}^{N} \mathbb{E}_{\theta \sim \mu} \big[ q(x_n; \theta) \big] - \delta,$$

i.e., the statement "the model predicts positive outcomes for males on average at least as much as for the whole population." In contrast, accommodating statistical parity for females requires considerable deviations from the reference model $\pi$ ($\lambda_{\text{female}}^\star \approx 160$). Without recalculating $\mu^\star$, we therefore know that even small changes in the tolerance $\delta$ for the female constraint would substantially change the

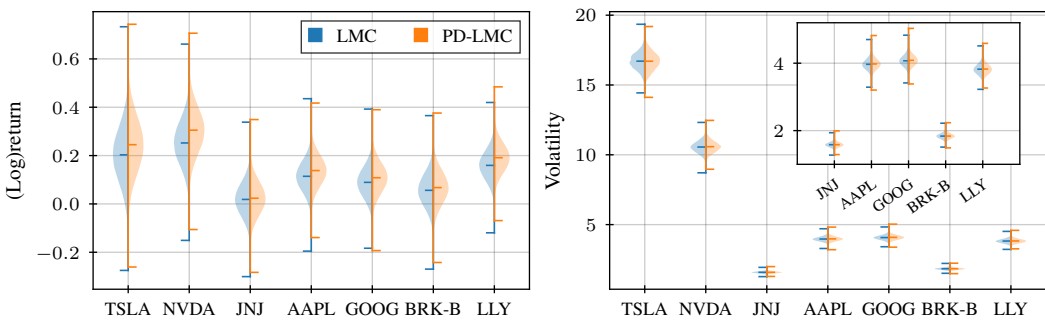

Figure 12: Counterfactual sampling of the stock market under a 20% average return increase on each stock: mean ($\rho$) and variance [$\mathrm{diag}(\Sigma)$] distributions.

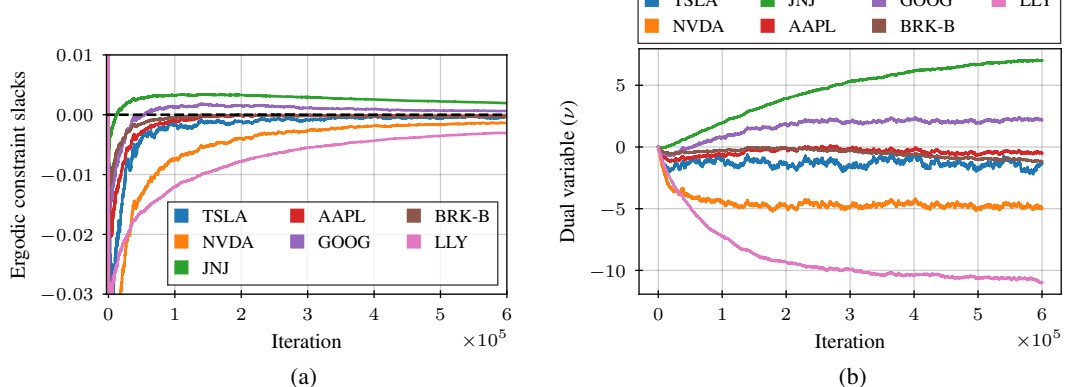

Figure 13: Counterfactual sampling of the stock market under a 20% average return increase on each stock: (a) ergodic average of constraint functions (slacks) and (b) dual variables.

distribution of outcomes. This statement is confirmed in Fig. 11b. Notice that this is only possible due to the primal-dual nature of PD-LMC.

### E.3.1 Stock market model

Counterfactual analyses based on the dual variables become more powerful as the number of constraints grow. To see this is the case, consider the *Bayesian stock market* model introduced in Section 2.2. Here, $\pi$ denotes the posterior model for the (log-)returns of 7 assets (TSLA, NVDA, JNJ, AAPL, GOOG, BRK-B, and LLY). The dataset is composed of 5 years of adjusted closing prices for a total of 1260 points per asset. The posterior is obtained by combining a Gaussian likelihood $\mathcal{N}(\rho, \Sigma)$ with Gaussian prior on the mean $\rho$ [$\mathcal{N}(0, 3I)$] and an inverse Wishart prior on the covariance $\Sigma$ (with parameters $\Psi = I$ and $\nu = 12$). Using the LMC algorithm ($\eta = 10^{-3}$; the last $3 \times 10^5$ samples are used after running $6 \times 10^5$ iterations), we collect samples from this posterior and estimate the mean and variance of the (log-)return for each stock (Table 3). In this case, $\Sigma$ is initialized to $10 \times I$.

We might now be interested in understanding what the market would look like if all stocks were to incur a 20% increase in their average (log-)returns. To do so, we use PD-LMC ($\eta_x = 10^{-3}$ and $\eta_\nu = 6 \times 10^{-3}$) to solve the following constrained sampling problem

$$\begin{aligned}
\underset{\mu \in \mathcal{P}_2(\mathbb{R}^d)}{\text{minimize}} \quad & \mathrm{KL}(\mu \| \pi) \\
\text{subject to} \quad & \mathbb{E}_{(\rho, \Sigma) \sim \mu}\big[\rho_i\big] = 1.2 \bar{\rho}_i, \quad i = 1, \dots, 7,
\end{aligned} \tag{PVIII}$$

where $\bar{\rho}_i$ is the mean (log-)return of the $i$-th stock shown in Table 3. The distribution of $\rho$ and the diagonal of $\Sigma$ are compared to those from the unconstrained model in Fig. 12. Notice that, though we only impose constraints on the average returns $\rho$, we also see small changes in the stock volatilities.

Inspecting the dual variables (Fig. 13b), we notice that three dual variables are essentially zero (TSLA, AAPL, and BRK-B). This means that their increased returns are completely dictated by those of other

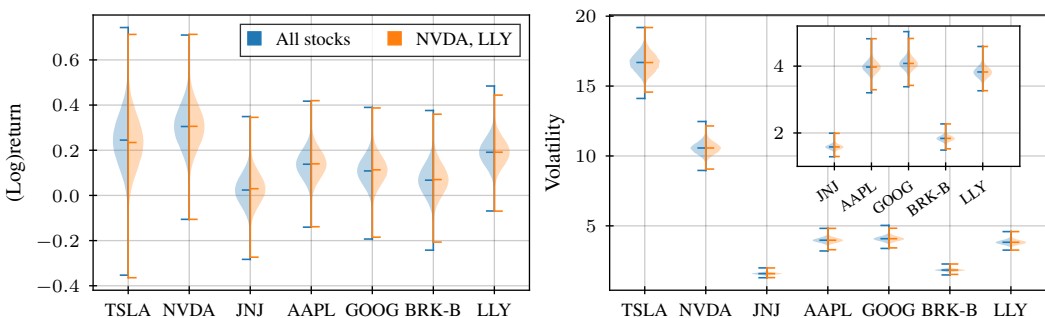

Figure 14: Distribution of mean ($\rho$) and variance [$\mathrm{diag}(\Sigma)$] of the stock market constrained to a 20% average return increase on all stocks vs. only LLY and NVDA.

Table 3: Mean $\pm$ standard deviation of mean (log-)returns ($\rho$).

|  | Reference model ($\pi$) | +20% (all stocks) | +20% (LLY and NVDA) |
|---|---|---|---|
| TSLA | $0.20 \pm 0.12$ | $0.24 \pm 0.12$ | $0.23 \pm 0.12$ |
| NVDA | $0.25 \pm 0.09$ | $0.31 \pm 0.09$ | $0.31 \pm 0.09$ |
| JNJ | $0.02 \pm 0.07$ | $0.02 \pm 0.07$ | $0.03 \pm 0.07$ |
| AAPL | $0.11 \pm 0.06$ | $0.14 \pm 0.06$ | $0.14 \pm 0.06$ |
| GOOG | $0.09 \pm 0.06$ | $0.11 \pm 0.06$ | $0.11 \pm 0.06$ |
| BRK-B | $0.06 \pm 0.07$ | $0.07 \pm 0.07$ | $0.07 \pm 0.07$ |
| LLY | $0.16 \pm 0.06$ | $0.19 \pm 0.06$ | $0.19 \pm 0.06$ |

stocks. Said differently, their returns increasing 20% is consistent with the reference model $\pi$ *conditioned* on the other returns increasing. Proceeding, two stocks have negative dual variables (LLY and NVDA). This implies that bringing their constraints *down to* $\bar{\rho}_i$ would yield a *decrease* in $P^\star$ (distance to the reference model $\pi$). This is in contrast to JNJ and GOOG, whose positive $\lambda$'s imply that we should should *increase* their returns to reduce $P^\star$. Indeed, by inspecting the ergodic slacks (Fig. 13a) we see that all stocks approach zero (i.e., feasibility), but that JNJ and GOOG do so from above. This behavior is expected according to Prop. 3.4.

These observations show two things. First, that an increase in the average returns of LLY and NVDA is enough to drive up the returns of all other stocks. In fact, it leads to essentially the same distribution as if we had required the increase to affect all stocks (Fig. 14). Second, that the increase we would see in JNJ (and to a lesser extent GOOG) would actually be larger than 20%. Once again, we reach these conclusion without any additional computation. Their accuracy can be corroborated by the results in Table 3.

# F  Example application of Prop. 2.2

In this section we illustrate the result in Prop. 2.2, i.e., we show that given solutions $(\lambda^\star, \nu^\star)$ of (DI), the constrained sampling problem (PI) reduces to sampling from $\mu_{\lambda^\star \nu^\star} \propto e^{-U(\cdot, \lambda^\star, \nu^\star)}$.

Indeed, consider a standard Gaussian target, i.e., $\pi \propto e^{-\|x\|^2/2}$, and the linear moment constraint $\mathbb{E}[x] = b$, for $b \in \mathbb{R}^d$. This can be cast as (PI) with $f(x) = \|x\|^2/2$ and $h(x) = b - x$ (no inequality constraints, i.e., $I = 0$). Clearly, the solution of (PI) in this case is $\mu^\star = \mathcal{N}(b, I)$, i.e., a Gaussian distribution with mean $b$. What Prop 2.2 claims is that rather than directly solving (PI), we can solve (DI) to obtain a Lagrange multiplier $\nu^\star$ such that $\mu^\star = \mu_{\nu^\star}$ for $\mu_\nu$ defined as in (4).

In this setting, we can see this is the case by doing the computations explicitly. Indeed, we have

$$\mu_\nu(x) \propto \pi(x) e^{-\nu^\top h(x)} = \exp\left[ -\frac{\|x\|^2}{2} - \nu^\top(b - x) \right].$$

Completing the squares, we then obtain

$$\mu_\nu(x) \propto \exp\left[ -\frac{\|x - \nu\|^2}{2} + \frac{\|\nu\|^2}{2} - \nu^\top b \right]. \tag{49}$$

To compute the Lagrange multiplier $\nu^\star$, notice from the definition of the dual function in (6) that the dual problem (DI) is in fact a ratio of normalizing factors. Explicitly,

$$\nu^\star = \operatorname*{argmax}_{\nu \in \mathbb{R}^d} \, \log\left( \frac{\int \pi(x) dx}{\int \mu_\nu(x) dx} \right) = \operatorname*{argmax}_{\nu \in \mathbb{R}^d} \, \log\left[ \frac{\int \exp\left( -\frac{\|x\|^2}{2} \right) dx}{\exp(\|\nu\|^2/2 - \nu^\top b) \int \exp\left( -\frac{\|x - \nu\|^2}{2} \right) dx} \right].$$

Immediately, we obtain

$$\nu^\star = \operatorname*{argmax}_{\nu \in \mathbb{R}^d} \, -\|\nu\|^2/2 + \nu^\top b = b. \tag{50}$$

Note that the dual problem is a concave program, as is always the case [30]. To conclude, we can combine (49) and (50) to get

$$\mu_{\nu^\star}(x) \Big|_{\nu^\star = b} \propto \exp\left( -\frac{\|x - b\|^2}{2} - \frac{\|b\|^2}{2} \right) \Rightarrow \mu_{\nu^\star} = \mathcal{N}(b, I) = \mu^\star.$$

The main advantage of using Algorithms 1 and 2 is that we do not need to determine the Lagrange multipliers $(\lambda^\star, \nu^\star)$ to then sample from $\mu_{\lambda^\star \nu^\star} = \mu^\star$. Indeed, Theorems 3.3 and 3.6 show that these stochastic primal-dual methods do both things simultaneously, without explicitly evaluating any expectations.

