# OpenReview forum: "Constrained Sampling with Primal-Dual Langevin Monte Carlo"
_NeurIPS.cc/2024/Conference — NeurIPS 2024 poster_

### Official Review · Reviewer_ZMkL · 2024-07-08

**Soundness:** 2
**Presentation:** 3
**Contribution:** 2
**Rating:** 5
**Confidence:** 3

**Summary:**

This paper aims to solve the constrained sampling problem via primal-dual method. The authors proposed a new sampling method PD-LMC and provide detailed convergence analysis. Several numerical experiments were conducted to verify the sampling method.

**Strengths:**

The structure of paper is easy to follow. The authors discussed detailed background and related work. The authors also established solid convergece guarantee for their algorithms under different settings.

**Weaknesses:**

1. The primal-dual method for constrained sampling problem is not novel. Similar algorithms were also studied in [1]. I notice that PD Langevin in [1] requires the estimation of expectation with respect to sampling distribution. But it can easily addressed by running many particles in parallel. Therefore I take the theoretical analysis as the most important contribution of this paper. The weaknesses are listed as following.

2. In the theoretical analysis part, the authors didn't show the rate of violation of constraints, i.e. if $\mu_k$ satisfies constraints. And this leads to the following one.

3. In Theorem 3.3, when there are equality constraints, $\mu^*$ is supported on a low dimension manifold while $\mu_k$ is not, then $KL(\mu_k\\|\mu^*)$ is not well defined. Hence I doubt the correctness of Theorem 3.3. I suggest the authors should consider metric not depending on density ratio, like W2 distance or TV distance.

4. In the numerical experiments, the paper lacks comparison with other methods, e.g. mirror Langevin related methods for sampling on convex set; PD Langevin and Control Langevin in [1] in rate-constrained Bayesian models.

[1] Liu, Xingchao, Xin Tong, and Qiang Liu. "Sampling with trusthworthy constraints: A variational gradient framework." Advances in Neural Information Processing Systems 34 (2021): 23557-23568.

**Questions:**

Please see weakness part.

**Limitations:**

The limitations are pointed out in the paper.

---

> ### Author Rebuttal · Authors · 2024-08-05
>
> We thank the reviewer for their positive comments on our results, particularly our theoretical guarantees. As the reviewer correctly notes, in contrast to [22] ([1] in the reviewer's comments), our analysis handles two levels of approximation: time- and space-discretization. We next address their questions.
>
> **Q1** Indeed, as we clearly acknowledge in the paper, Algorithms 1 and 2 can be seen as approximations of the PD Langevin from [22]. But there are substantial computational and theoretical differences between these methods.
>
> The dual variable updates in the algorithms from [22] require computing an expectation over the current particle distribution [see (10)], which is intractable. While we agree that it can be *approximated* by running many particles in parallel, this would result in a substantially higher computational cost than the stochastic updates in our paper. Indeed, we show PD-LMC (Algorithm 1) converges using a single-particle approximation.
>
> Furthermore, it is not clear from [22] how these approximation errors impact the convergence of the algorithms since it only provides guarantees for exact dual updates (computing expectations). In contrast, our theoretical analyses exactly characterize the bias introduced by approximation errors (Proposition D.1) and show that they do not affect the convergence in the convex setting (Theorem 3.3). This is confirmed by our experiments. These results hold for discrete-time, expectation-free algorithms.
>
> That being said, we acknowledge that we could have better illustrated the relative performance of PD Langevin and our stochastic version PD-LMC. We have addressed this oversight and provided an illustration of the performance of both schemes as the number of LMC particles $N$ grows for the one-dimensional truncated Gaussian and the rate-constrained Bayesian model (see Fig. 1 and 2 in the pdf attached to the global response). This hopefully illustrates that one can achieve essentially the same sampling performance as PD Langevin at a considerably lower computational cost.
>
> **Q2** The reviewer has a point: our results do not *directly* analyze the decrease rate of the constraint violation. Yet, our theoretical guarantees show that PD-LMC converges to $\mu^\star$ that is feasible by definition. Hence, Theorems 3.3 and 3.5 do address the issue of feasibility. Additionally, it is possible to use our results to explicitly control the constraints violation along iterations.
>
> Consider the results in Theorem 3.3, namely (13):
> $$\frac{1}{K} \sum\_{k = 1}^K \mathrm{KL}(\mu\_{k} \| \mu^\star)+ \frac{m}{2} W\_2^2(\mu\_k,\mu^\star)\leq\frac{R\_0^2}{\eta K}+ \eta G^2 \bigg( 3 +\sum\_{k = 1}^K \frac{\mathbb{E}\_{\lambda}[\|\lambda\_k\|^2]+ \mathbb{E}\_{\nu}[\|\nu\_k\|^2]}{K}\bigg) \triangleq \Delta\_K.$$
> Since $x\_k \sim \mu\_k$ and $\mathbb{E}\_{\mu^\star}[g] \leq 0$, we obtain that
> $$\mathbb{E} \bigg[ \frac{1}{K} \sum\_{k=1}^K g(x\_k) \bigg]\leq \frac{1}{K} \sum\_{k=1}^K \int g d\mu\_k - \int g d\mu^\star\leq \frac{1}{K} \sum\_{k=1}^K \Big| \int g d\mu\_k - \int g d\mu^\star \Big|\leq \sqrt{\frac{1}{K} \sum\_{k=1}^K \Big| \int g d\mu\_k - \int g d\mu^\star \Big|^2},$$
> where we used the $\ell\_1/\ell\_2$-norm relation.
>
> If $g$ is bounded by 1, the summands are bounded by $TV(\mu\_k,\mu^\star)^2$. Using Pinsker's inequality, we therefore obtain
> $$\mathbb{E} \bigg[ \frac{1}{K} \sum\_{k=1}^K g(x\_k) \bigg]\leq \sqrt{\frac{1}{2K} \sum\_{k=1}^K \mathrm{KL}(\mu\_{k} \| \mu^\star)^2}\leq \sqrt{\frac{\Delta\_K}{2}}$$
> If $g$ is $1$-Lipschitz (and $f$ is $m$-strongly convex), we have
> $$\Big| \int g d\mu\_k - \int g d\mu^\star \Big|^2\le W\_1^2(\mu\_k,\mu^\star) \le W\_2^2(\mu\_k, \mu^\star),$$
> which implies
> $$\mathbb{E} \bigg[ \frac{1}{K} \sum\_{k=1}^K g(x\_k) \bigg]\leq \sqrt{\frac{2 \Delta\_K}{m}}.$$
> The same arguments hold for $h$.
>
> Hence, our results can be used to control (in expectation) the constraint violation of ergodic averages along the PD-LMC trajectories. These behaviors are also observed in our experiments. For instance, Fig. 3 of the pdf attached to the global response shows the ergodic constraint slacks $\sum h(x\_k)/K$ for the equality constraints of the Bayesian stock market problem (Fig. 2 in the paper). Notice that by the end of the simulation their values are within $3 \times 10^{-3}$ of zero. We thank the reviewer for raising this point and will include these corollaries and discussions in the camera-ready version.
>
> **Q3** We respectfully, but strongly disagree with this statement. We believe there may have been a misunderstanding arising from the fact that the constraints in (PI) are *distribution constraints* rather than *constraints on the support of $\mu$*. We refer the reviewer to Section 2.2 for a detailed discussion of the differences between these constraint types. In our case, imposing a moment constraint (even as an equality constraint) does not result in $\mu^\star$ being supported on a lower-dimensional manifold. Another way to see this is by noticing that $\mu^\star \propto \pi \exp^{-\nu^\star h}$ [see Prop. 2.2(iv)]. Hence, for $\pi$ full-dimensional and $(\nu^\star, h)$ finite [which is the case, see Prop. 2.2(ii)], $\mu^\star$ will also be full-dimensional. For a concrete example, we refer the reviewer to the response to *Q2 of reviewer 4nfu*, where we derive an explicit solution for (DI) under an equality constraint.
>
> **Q4** We have included results for the Mirrored Langevin in the two-dimensional truncated Gaussian as well as the PD Langevin from [22] for the rate-constrained Bayesian problem (Figs. 4 and 2 respectively in the pdf attached to the global response). We note again that the latter requires an explicit integration that is intractable and that using LMC chains to approximate it reduces to the settings of Algorithms 1-2, whose convergence is guaranteed by our results (Theorems 3.3 and 3.5). As expected, we therefore obtain the same results albeit at a considerable increase in computational complexity.

---

> > ### Comment · Reviewer_ZMkL · 2024-08-08
> >
> > Thanks for the feedback. Most of my concerns are addressed and I'm happy to raise the score. The rate of violation of constraints is an important proposition that should be added to the main text. I would like to mention that the proposed PD LMC algorithm is the same with PD Langevin in [22] when particle number $N=1$, while the theoretical analysis in this paper is solid and novel.

---

### Official Review · Reviewer_4nfu · 2024-07-08

**Soundness:** 3
**Presentation:** 4
**Contribution:** 3
**Rating:** 7
**Confidence:** 5

**Summary:**

The paper studies constrained sampling schemes. The objective function is the KL divergences with a target distribution, while the constraint set is given as some expectation equations or inequalities. The authors rewrite the problem into a saddle point formulation and study the Wasserstein gradient descent and L2 gradient ascent directions. This results in a particular Langevin dynamics with a dual ascent direction. The authors study the convergence analysis of the proposed algorithm. Several numerical examples demonstrate the effectiveness of the proposed method.

**Strengths:**

The authors clearly explain the constrained optimization problems. They apply the primal-dual gradient descent-ascent algorithm based on the Wasserstein space to solve the optimization problem. Convergence analysis is presented using tools in optimal transport.

Sampling from a convex set is a very good application.

**Weaknesses:**

There is a lack of literature on generalized optimization problems in Wasserstein spaces.

Wang et al. Accelerated Information Gradient flow. Journal of Scientific Computing. 2022.

Wang et al. Information Newton's flow: second-order optimization method in probability space, 2020.

Tan et al. Noise-Free Sampling Algorithms via Regularized Wasserstein Proximals, 2023.

**Questions:**

1. For the time update, the authors perform the forward Euler time step for the primal and dual variables.  Do the authors expect some advantages if one applies the proximal steps?  What is the main difference in analyzing the primal-dual algorithm in Wasserstein space and the classical Euclidean space.

2. Can the authors provide some simple examples, such as Gaussian target distributions and linear moment constraints, to explain the main result? This could partially answer the first question.

**Limitations:**

There are no limitations.

---

> ### Author Rebuttal · Authors · 2024-08-05
>
> We thank the reviewer for their enthusiastic opinion of our paper. We next address their questions one-by-one.
>
> **Weakness** The reviewer has a point. We focused on regular Langevin dynamics and its literature, but this paper is indeed part of the large line of work of sampling as an optimization in the space of probability measures. It is therefore connected (or could be extended) to many other schemes. We appreciate the suggested references and will include them in the manuscript.
>
> **Q1.1** Since our main focus was on deriving and studying a computationally-friendly algorithm for the constrained sampling problem (PI), we did not consider proximal updates that are more expensive than the fully explicit Euler updates used in this paper (for both primal and dual variables). However, we acknowledge that investigating proximal steps is an interesting future direction and we believe that it could in fact lead to stronger theoretical guarantees (under additional assumptions) as is the case for gradient descent-ascent in Euclidean space (see, e.g., [40]). We will make note of this point in the revised version of the paper.
>
> **Q1.2** To a large extent, our approach is akin to that used to study any optimization algorithm/dynamical system: we construct a Lyapunov function to bound the duality gap $L(\mu,\lambda^\star,\nu^\star) - L(\mu^\star, \lambda, \nu)$ [note from the saddle-point relation (8) that this is the right optimality measure]. The techniques used to bound this gap in Wasserstein space, however, are different.
>
> Indeed, while the dual (DI) remains a (finite dimensional, non-smooth) optimization problem in Euclidean space, the primal (PI) is an (infinite dimensional, smooth) optimization problem in Wasserstein space. The proof must therefore account for the mixed nature of the Lagrangian. Additionally, Algorithm 1 performs stochastic updates in both the primal (step 3) and dual (steps 4-5). Hence, the potential $U$ used to update $x$ in step 3 is a random variable and we have to deal with the joint distribution $(x\_k,\lambda\_k,\nu\_k)$. This is an important distinction with the traditional LMC or when the dual updates (steps 4-5) are performed using exact expectations as in [22]. To address these issues, we instead use conditional laws to obtain our guarantees in expectation over realizations of $(\lambda\_k,\nu\_k)$.
>
> **Q2** Consider a standard Gaussian target, i.e., $\pi \propto e^{-\\|x\\|^2/2}$, and the linear moment constraint $\mathbb{E}[x] = b$, for $b \in \mathbb{R}^d$. This can be cast as (PI) with $f(x) = \\|x\\|^2/2$ and $h(x) = b - x$ (no inequality constraints, i.e., $I = 0$). Clearly, the solution of (PI) in this case is $\mu^\star = \mathcal{N}(b,I)$. What Prop 2.2 claims is that rather than directly solving (PI), we can solve (DI) to obtain a Lagrange multiplier $\nu^\star$ such that $\mu^\star = \mu\_{\nu^\star}$ for $\mu\_{\nu}$ defined as in (5) (line 134).
>
> We can show this indeed the case here by doing computations explicitly. Indeed, we have
> $$
> \mu^\star(x) = \mu\_{\nu^\star}(x) \propto \pi(x) e^{-(\nu^\star)^\top h(x)}
>     = \exp\big[ -\\|x\\|^2/2 -(\nu^\star)^\top (b-x) \big].
> $$
> Completing the squares we obtain
> $$
> \mu^\star(x) \propto \exp\big[ -\\|x - \nu^\star\\|^2/2 + \\|\nu^\star\\|^2/2 -(\nu^\star)^\top b \big].
> $$
> From the definition of the dual function in (7) (line 139), we can write (DI) explicitly as a ratio of normalizing factors, namely
> $$
> \nu^\star
>     = \text{argmax}\_{\nu}\ \log\Big( \frac{\int \pi(x)dx}{\int \mu\_\nu(x) dx} \Big)
>     = \text{argmax}\_{\nu}\ \log\Big( \frac{\int \pi(x)dx}{\exp(\\|\nu\\|^2/2 -\nu^\top b) \int \pi(x) dx} \Big).
> $$
> Immediately, we obtain
> $$
> \nu^\star = \text{argmax}\_{\nu}\ -\\|\nu\\|^2/2 +\nu^\top b = b.
> $$
> Note that, as we mention in the text, the dual problem is indeed a concave program. To conclude, observe that as we indeed have
> $$
> \mu^\star(x) = \mu\_{\nu^\star}(x) \Bigm\vert\_{\nu^\star = b} \propto \exp\big[ -\\|x - b\\|^2/2 - \\|b\\|^2/2 \big]
> \text{, i.e., $\mu^\star(x) = \mathcal{N}(b,I)$.}
> $$
>
> Our main results (Theorems 3.3 and 3.5) show that it is not necessary to first determine the Lagrange multipliers $(\lambda^\star,\nu^\star)$ to then sample from $\mu\_{\lambda^\star\nu^\star} = \mu^\star$. Indeed, the stochastic primal-dual Langevin methods in Algorithms 1--2 simultaneously determine $(\lambda^\star,\nu^\star)$ and sample from $\mu\_{\lambda^\star\nu^\star} = \mu^\star$. Additionally, they do so without explicitly evaluating any expectations.
>
> Please do not hesitate to let us know if something is not clear. If this explanation indeed clarifies some of our duality derivations, we consider including it in the appendix as a warm-up example.

---

> > ### Comment · Reviewer_4nfu · 2024-08-08
> > **Reply to authors**
> >
> > The authors carefully address my questions. I also like the example of Gaussian distributions. I suggest the publication of this paper.

---

### Official Review · Reviewer_d9K2 · 2024-07-11

**Soundness:** 3
**Presentation:** 3
**Contribution:** 2
**Rating:** 5
**Confidence:** 4

**Summary:**

In this work, the focus is on a constrained optimisation problem in the space of measures. Specifically, the goal is to obtain a distribution / samples from a distribution which is close in KL to a target distribution while also satisfying a set of statistical constraints. The paper discusses this somewhat atypical problem, and designs the PD-LMC method to generate approximate sample from such a distribution. Theoretical properties of PD-LMC are also presented, along with some numerical experiments to showcase the working of this method.

**Strengths:**

### Quality and Clarity

The paper reads well, with a clear description of the objective at hand and the necessary background to help the reader grasp the problem and the proposed solution. For someone used to the classical constrained sampling problem, the examples in Section 2.2 definitely helped contextualize the problem better, and is appreciated. The dual formulation which is central to this work is also adequately presented, which helps understanding the algorithm. It is interesting that the proposed method gets away without computing expectations and replacing them in a single particle.

### Originality and Significance

The idea is novel, and looks like a clean way to adapt ideas from Euclidean optimization for the distributional constrained problem considered here. I cannot quite comment on the significance of this proposed method as I'm not familiar with this flavor of constrained sampling. Judging by the numerical experiments in section 4, the method appears to work well in low-dimensional setting.

**Weaknesses:**

The method is interesting and novel, but there is some loss of intuition for me, which I've tried to express as a question in Q2. Essentially, the proposed method performs two levels of approximations which while practically appealing, makes it hard to follow. I also have a minor issue with laws of $x_{k}$ and the expectations appearing in Theorem 3.3. Essentially, the concern is how there's no stochasticity on the LHS, and this is where my confusion with working with laws and samples is confusing to me.

**Questions:**

1. Is the purpose of the $x$-updates simply to approximate the discretisation in the steps 11b and 11c? This was particularly confusing to me because problem DI doesn't involve any $\mu$, which is what is sought out to be solved.
2. Following-up on the previous question, to understand better, the line of reasoning is: Eq. 9 would be what is ideal, but this requires knowing $\mu_{\lambda_{k}, \nu_{k}}$ and expectations under this. So, [22] proposed Eq. 10 in the equality constrainted setting, which is extended to Alg 1 in this paper. Since $\mu_{\lambda_{k}, \nu_{k}}$ is a tilted version of $\pi$, what would a proposal like approximately computing the expectations in Eq. 9 using a high-accuracy sampler like MALA do (notwithstanding the sample waste)? I suppose something similar is being done in Algorithm 2?
3. What is preventing the dual variables from blowing up in magnitude? In other words, how large can $\max_{k} \mathbb{E}[\|\lambda_{k}\|^{2}] + \mathbb{E}[\|\nu_{k}\|^{2}]$ be / what rate does it grow at?
4. Why are equality constraints disregarded for PD-LMC with LSI potentials? Since $\nu$ doesn't exist, why does it show up in Algorithm 2?
5. Can you give a concrete example for when Assumption 3.4 is satisfied, i.e., with $g$ being a bounded perturbation, and a bound on $\sigma$?

**Limitations:**

Not quite, it would be nice if the authors could comment on some limitations of their analysis / methods.

---

> ### Author Rebuttal · Authors · 2024-08-05
>
> We thank the reviewer for their positive comments on our paper.
>
> **Weakness** The reviewer has a point that to obtain a practical algorithm we perform stochastic updates in both the primal (step 3) and dual (steps 4-5), which complicates things. This in fact marks an important distinction with the analyses of the traditional LMC or when the dual updates are performed using exact expectations as in [22], in that we now have to handle the full joint distribution $(x,\lambda,\nu)$. To overcome this challenge, we work with conditional laws to obtain "in expectation" guarantees. That is why the bounds in Theorem 3.3 are not stochastic.
>
> Indeed, note that (13) comes from the inequality (line 617, Appendix C)
> $$\frac{1}{K} \sum\_{k = 1}^K \mathbb{E}\_{\lambda,\nu} \Big[\text{KL}(\tilde{\mu}\_{k} \| \mu^\star) + \frac{m}{2} W\_2^2(\tilde{\mu}\_k,\mu^\star)\Big] \leq\frac{R\_0^2}{\eta K}+ \eta G^2 \bigg( 3 +\sum\_{k = 1}^K \frac{\mathbb{E}\_{\lambda}[\|\lambda\_k\|^2]+ \mathbb{E}\_{\nu}[\|\nu\_k\|^2]}{K}\bigg),$$
> where $\tilde{\mu}\_k$ is the conditional law of $x\_k \vert \\{\lambda\_{\ell}, \nu\_{\ell}\\}\_{\ell<k}$. Our proof analyzes the evolution of this conditional law [as in (18)] and then apply Jensen's inequality using the fact that $\mathbb{E}\_{\lambda,\nu} [\tilde{\mu}\_k] = \mu\_k$, the law of $x\_k$ (line 618). Due to space constraints, we did not include this technical discussion in the main text but we plan to do so in the final version.
>
> **Q1** Note that (DI) involves $\mu$ through its objective, the dual function $d$ from (7). But the reviewer is correct that, in contrast to (PI) that directly seeks $\mu^\star$, the dual problem (DI) seeks $(\lambda^\star,\nu^\star)$ that can then be used to obtain $\mu^\star$ by $\mu\_{\lambda^\star\nu^\star}$ defined in (5)--(6) (line 134). In fact, finding $\mu^\star$ and $(\lambda^\star,\nu^\star)$ are equivalent [Prop. 2.2(iv)]. Hence, the goal of the $x$-update is in fact twofold: sampling from $\mu\_{\lambda^\star\nu^\star}$, i.e., $\mu^\star$, and compute the $(\lambda^\star,\nu^\star)$ [using the discrete versions of (11b)--(11c) in steps 4-5 of Algorithm 1].
>
> **Q2** The reviewer is correct. Computationally, (9) is intractable; (10) proposed by [22] is better, but still intractable (due to the expectations); but Algorithm 1 is practical and can be implemented without approximations. From the point-of-view of convergence, (9) performs "dual ascent" and would converge to $(\lambda^\star,\nu^\star)$; the continuous-time counterpart of (10) was shown to converge in [22] (we extend these results in Theorem 3.3 by showing the discrete version itself converges under milder conditions on $g$, as we discuss in line 247); and we prove that Algorithm 1 also converges (again, under milder assumptions).
>
> The reviewer's proposal would indeed yield Algorithm 2 (with an extra Metropolis acceptance step). But while this approach has advantages in the non-convex case (Section 3.2), its benefits do not necessarily outweigh the increased computational cost. For instance, consider the sample mean estimate of the one-dimensional truncated Gaussian from Fig. 1 (main paper) as a function of the number of LMC iterations (step 3) taken in Algorithm 1 (see Fig. 1 in the pdf attached to the global response). I.e., for $N=1$ we have Algorithm 1 and for $N > 1$ we follow the reviewer's proposal. Note that the additional computational cost does not improve convergence in this case (especially when considering the number of "LMC steps" $N$ which is proportional to the computational complexity).
>
> **Q3** This is hard to guarantee in general, even in the Euclidean convex case (see, e.g., [35]). As we mention in the paper (line 235), a common solution is to clip the iterates $(\lambda\_k,\nu\_k)$ in Algorithm 1 by some upper bound on $(\lambda^\star,\nu^\star)$. Such bounds are well-known and can be obtained under a variety of constraint qualifications such as Assumption 2.1, although explicit bounds for equality constraints require additional assumptions (see, e.g., [26,35] or the more general [Gauvin, "A necessary and sufficient regularity condition to have bounded multipliers in nonconvex programming," 1977]). Alternatively, a decreasing "weight decay" regularization can be used (see, e.g., [62]). In some cases, it is even possible to directly bound the iterates $\lambda\_k$ (see Lemma D.3). For these reasons, we chose to present our results in the form of Theorem 3.3, which we find to be more applicable, and then comment on potential solutions after the theorem. We will expand our remark on this point in the camera-ready version.
>
> **Q4** The reviewer is correct: there is a typo in step 4 of Algorithm 2. We omit the equality constraints in the non-convex case to keep the analysis manageable. Deriving results similar to Lemma D.3 and D.4 for equality constraints requires additional assumptions (see response to Q3) and substantial derivations that we felt obscured the results. We note that it is sometimes possible to cast equality constraints as two tight inequalities, although this might lead to numerical issues. We will expand on this remark in the camera-ready version.
>
> **Q5** Let $f(x) = x^2/2$ ($1$-strongly convex) and $g(x) = \sin(x)$ (bounded). We therefore have $\mu\_\lambda \propto e^{-x^2/2-\lambda\sin(x)}$. For $\lambda = 0$, Assumption 3.4 holds with $\sigma = 1$. As $\lambda$ increases, $\sigma$ decreases but $\sigma > 0$ for all finite $\lambda$. In fact, $\sigma \geq e^{-2\lambda}$ (see, e.g., [42, Prop. 5.1.6] or Theorem 1.1 in [Cattiaux and Guillin, "Functional inequalities for perturbed measures with applications to log-concave measures and to some Bayesian problems," 2022]). Since $\lambda^\star$ is finite (see bound in Lemma D.3), we could explicitly clip the iterates $\lambda\_k$ and get an *a priori* bound on $\sigma$. But even without any explicit limit, we show in Theorem 3.5 that $\mathbb{E} \\| \lambda\_k \\|\_1$ is bounded for all $k$.

---

> > ### Comment · Reviewer_d9K2 · 2024-08-10
> > **Thank you for your rebuttal**
> >
> > Q3: I feel like this is an important point to cover. Specifically if the quantity increases faster than $K$, then the analysis would yield a vacuous bound. In your method, 1) how would you obtain a bound, 2) how would you enforce a bound on these variables, and 3) how would this affect the guarantees for the method?
> >
> > I have no further questions aside from the ones above.

---

> > > ### Author Response · Authors · 2024-08-12
> > >
> > > We agree with the reviewer that these are important points and address them in the discussion after Theorem 3.3 in the current version of the manuscript (lines 235-244). There, we argue for the form of Theorem 3.3 rather than modifying Algorithm 1 because Theorem 3.3 also shows there exists a sequence of step sizes such that $(\lambda\_k,\nu\_k)$ are bounded. Additionally, none of our experiments enforce bounds (they implement PD-LMC exactly as described in Algorithm 1). Yet, we always observe the dual variables converging (see, e.g., Fig. 2 in the manuscript), suggesting that pathological cases are not common. That being said, the theory would not be complete without addressing these cases, which we discuss in the remarks of lines 235-244. We expand our treatment of these points, including the discussion below, in the revised version of the manuscript. If any point remains unclear, we are happy to provide further details.
> > >
> > > To the reviewer's points:
> > >
> > > 1) **Bounds on $(\lambda^\star,\nu^\star)$**: Under Assumption 2.1, we provide such a bound on $\\|\lambda^\star\\|\_1$ in Lemma D.3. Explicitly, suppose there exists a $\mu^\dagger$ such that $\mathrm{KL}(\mu^\dagger \\| \pi) \leq C$, $\mathbb{E}\_{\mu^\dagger} [g\_i] \leq -\delta < 0$, and $\mathbb{E}\_{\mu^\dagger} [h\_j] = 0$ (Assumption 2.1). Then, by definition
> > >     $$
> > >     D^\star = d(\lambda^\star,\nu^\star) \leq \mathrm{KL}(\mu^\dagger \\| \pi) + \sum\_i \lambda^\star\_i \mathbb{E}\_{\mu^\dagger} [g\_i] + \sum\_j \nu^\star\_j \mathbb{E}\_{\mu^\dagger} [h\_j] \leq C - \\|\lambda^\star\\|\_1 \delta
> > >     $$
> > >     for the dual function in (7). By strong duality, $D^\star = P^\star \geq 0$, which yields
> > >     $$
> > >     \\|\lambda^\star\\|\_1 \leq \frac{C}{\delta}.
> > >     $$
> > >     A bound on $\nu^\star$, though more complicated, can be obtained under a similar constraint qualification (see, e.g., the general [Gauvin, "A necessary and sufficient regularity condition to have bounded multipliers in nonconvex programming," 1977]).
> > >
> > > 2) **Enforcing the bound**: Suppose $\\|\lambda^\star\\|\_1,\\|\nu^\star\\|\_1 \leq B$. Suffices it to replace steps 4 and 5 in Algorithm 1 by
> > >     $$
> > >     \lambda\_{k+1} = \Big[ \lambda\_k + \eta\_k g(x\_k) \Big]\_0^{B}
> > >     \quad \text{and} \quad
> > >     \nu\_{k+1} = \Big[ \nu\_k + \eta\_k h(x\_k) \Big]\_{-B}^{B},
> > >     $$
> > >     where $[z]\_{L}^U = \min(\max(z,L),U)$. This is a common solution used in the Euclidean case (see, e.g., [35]).
> > >
> > > 3) **How does this affect the guarantees?** It does not. Indeed, note that proof of Theorem 3.3 is based on the fact that (17) is a Lyapunov function. This is proved in Lemma C.1. When bounding the Euclidean norm of the dual variables (line 642), the first step is realizing that the projection $\lambda \mapsto [\lambda]\_0^B$ and $\nu \mapsto [\nu]\_{-B}^B$ are contractions, since $\lambda^\star\_i \in [0,B]$ and $\nu^\star\_j \in [-B,B]$. Hence, we can use
> > >     $$
> > >     \\| [\lambda]\_0^B - \lambda^\star \\|^2 \leq \\| \lambda - \lambda^\star \\|^2
> > >     \quad \text{and} \quad
> > >     \\| [\nu]\_{-B}^B - \nu^\star \\|^2 \leq \\| \nu - \nu^\star \\|^2
> > >     $$
> > >     to obtain (26). The proof then proceed as is and yields the exact same results as in Theorem 3.3, except that (13) [and similarly (14)] can now be rewritten as
> > >     $$
> > >         \frac{1}{K} \sum\_{k = 1}^K
> > >             \mathrm{KL}(\mu\_{k} \\| \mu^\star)
> > >             + \frac{m}{2} W\_2^2(\mu\_k,\mu^\star)
> > >         \leq
> > >         \frac{R\_0^2}{\eta K}
> > >         + \eta G^2 ( 3 + B^2 ).
> > >     $$

---

> > > > ### Comment · Reviewer_d9K2 · 2024-08-12
> > > > **Thank you for the further commentary**
> > > >
> > > > I have no further questions, and would like to maintain my score. I think that parts 2 and 3 from above would benefit the exposition.

---

### Author Rebuttal · Authors · 2024-08-05

We thank the editors and reviewers for their time and for the positive comments on our paper.

In the sequel, we respond to each of their questions individually. Throughout our responses, we refer to references and equations as numbered in the submitted version of the manuscript. We also refer to figures contained in the pdf attached to this response as "pdf attached to the global response."

---

### Comment · Area_Chair_oRkn · 2024-08-13
**Please comment on the relation of your work to ...**

Please comment on the relation of your work to ...

Salim and Richtarik, **Primal Dual Interpretation of the Proximal Stochastic Gradient Langevin Algorithm**, NeurIPS 2020,
https://papers.nips.cc/paper/2020/hash/2779fda014fbadb761f67dd708c1325e-Abstract.html

This seems relevant due to the primal-dual interpretation of Langevin described here. Moreover, the work directly addresses the issue of constrained sampling. This work is not cited, but seems very relevant. Is it not? If either case, I believe, this should be clearly explained here to the reviewers and in the paper.

Apologies for posting this so late; I hope you can still respond.

Reviewers: What do you think?

Best regards,

AC

---

> ### Comment · Reviewer_d9K2 · 2024-08-13
> **My understanding of the relation**
>
> I feel that the constrained setting in the submission deviates from the traditional constrained sampling setup in that the constraints are expressed as functionals, and the dual formulation extends the formulation in the Euclidean space to space of probability measures. The above paper appears more relevant to when the potential to sample from is a composite function (smooth and non-smooth parts). But I'm also interested to hear about the authors' thoughts (hopefully they can respond).

---

> ### Author Response · Authors · 2024-08-13
>
> This manuscript and [Salim and Richtarik] use primal-dual methods in distinct ways. As *reviewer d9K2* points out, they tackle the problem of sampling from composite potentials that can contain a non-smooth part restricting the support of the distribution. In contrast, we study (PI) which imposes statistical (moment) constraints on the distribution (see Table 1 and Section 2.2 for a detailed discussion on the differences between these constraint types). While (PI) can be used to tackle support constraints (as we show in Section 2.2 and illustrate in our experiments), it can also handle statistical restrictions for which proximal operators cannot be constructed (see examples in Section 2.2). Though our focus on computational aspects means we did not consider proximal updates in this paper, we do believe that studying proximal versions of PD-LMC is an interesting future direction (see more details in our response to *reviewer 4nfu*).
>
> Additionally, [Salim and Richtarik] consider stochastic "primal" updates [see (3) in their paper] as in *stochastic gradient Langevin* [33]. In constrast, PD-LMC (Algorithm 1) uses stochastic gradients only for its dual updates (steps 4-5). Indeed, note from (6) that $U$ is a deterministic function that depends only on parameters of the sampling problem (namely, $f,g,h$). Hence, its evaluation does not involve computing any expectation. Combining stochastic gradient Langevin with PD-LMC is also an interesting research direction but outside the scope of this work (see comment line 198).
>
> That being said, as with other papers mentioned in these reviews, [Salim and Richtarik] is certainly relevant related literature and we will definitely include these clarifications in the revised version of the manuscript.

---

### Decision · Program_Chairs · 2024-09-25

**Decision:**

Accept (poster)

**Comment:**

The focus of this paper is on a constrained optimization problem in the space of measures. The goal is to obtain samples from a distribution which is close in KL to a target distribution while also satisfying a set of statistical constraints. The paper discusses this somewhat atypical problem, and the authors design the PD-LMC method to generate an approximate sample. Theoretical properties of PD-LMC are studied. Numerical experiments showcase the workings of this method.

All reviewers were positive about this work (scores 5, 5, and 7), indicating that the strengths outweigh the weaknesses. Some of the strengths of the paper include:

- The paper is well written and reads well
- The objective of the research and the background help the reader grasp the problem and the proposed solution
- The dual formulation central to this work is also adequately presented
- The idea seems to be novel, and looks like a clean way to adapt ideas from Euclidean optimization for the distributional constrained problem considered here
- The authors establish solid convergence guarantee for their algorithms in various settings

I fully expect the authors to address all the observed weaknesses in the camera-ready version of the paper. On this condition, based on the above reasons, I propose the paper to be accepted.

AC